# A generic binding pocket for small molecule $I_{Ks}$ activators at the extracellular inter-subunit interface of KCNQ1 and KCNE1 channel complexes

**Magnus Chan[1†], Harutyun Sahakyan[2†‡], Jodene Eldstrom[1†], Daniel Sastre[1], Yundi Wang[1], Ying Dou[1], Marc Pourrier[1], Vitya Vardanyan[3]\*, David Fedida[1]\***

[1]Department of Anesthesiology, Pharmacology and Therapeutics, University of British Columbia, Vancouver, Canada; [2]Laboratory of Computational Modeling of Biological Processes, Institute of Molecular Biology, Yerevan, Armenia; [3]Molecular Neuroscience Group, Institute of Molecular Biology, Yerevan, Armenia

**\*For correspondence:**
ararat2025@gmail.com (VV);
david.fedida@ubc.ca (DF)

[†]These authors contributed equally to this work

**Present address:** [‡]National Center for Biotechnology Information, National Library of Medicine, National Institutes of Health, Bethesda, United States

**Competing interest:** The authors declare that no competing interests exist.

**Abstract** The cardiac $I_{Ks}$ ion channel comprises KCNQ1, calmodulin, and KCNE1 in a dodecameric complex which provides a repolarizing current reserve at higher heart rates and protects from arrhythmia syndromes that cause fainting and sudden death. Pharmacological activators of $I_{Ks}$ are therefore of interest both scientifically and therapeutically for treatment of $I_{Ks}$ loss-of-function disorders. One group of chemical activators are only active in the presence of the accessory KCNE1 subunit and here we investigate this phenomenon using molecular modeling techniques and mutagenesis scanning in mammalian cells. A generalized activator binding pocket is formed extracellularly by KCNE1, the domain-swapped S1 helices of one KCNQ1 subunit and the pore/turret region made up of two other KCNQ1 subunits. A few residues, including K41, A44 and Y46 in KCNE1, W323 in the KCNQ1 pore, and Y148 in the KCNQ1 S1 domain, appear critical for the binding of structurally diverse molecules, but in addition, molecular modeling studies suggest that induced fit by structurally different molecules underlies the generalized nature of the binding pocket. Activation of $I_{Ks}$ is enhanced by stabilization of the KCNQ1-S1/KCNE1/pore complex, which ultimately slows deactivation of the current, and promotes outward current summation at higher pulse rates. Our results provide a mechanistic explanation of enhanced $I_{Ks}$ currents by these activator compounds and provide a map for future design of more potent therapeutically useful molecules.

## eLife assessment

By combining electrophysiological analysis of mutant channels and molecular dynamics simulations, this **important** study identifies a common binding site for two structurally distinct activators of KCNQ1-KCNE1 channels. The findings represent an **important** advance for the field, with **convincing** functional and computational data to support the claims. The work will be of interest to those studying the binding of small molecule drugs to membrane protein complexes.

## Introduction

Potassium ion (K⁺) channel activators are important compounds in human health, as partial or complete loss of function of many K⁺ channels may lead to inherited or acquired diseases that have significant morbidity and mortality. The delayed cardiac rectifier potassium current, $I_{Ks}$, plays an important role, especially at high heart rates, in the physiological shortening of the cardiac action potential

(*Sanguinetti et al., 1996*). Unsurprisingly, mutations in both KCNQ1 (α-subunit) and KCNE1 (β-subunit), which when paired together give rise to the $I_{Ks}$ current (*Barhanin et al., 1996*; *Sanguinetti et al., 1996*; *Bendahhou et al., 2005*), have been implicated in cardiac arrhythmia syndromes such as long QT syndrome (LQTS) and atrial fibrillation (*Jervell and Lange-Nelsen, 1957*; *Wang et al., 1996*; *Chen et al., 2003*; *Eldstrom and Fedida, 2011*; *Olesen et al., 2014*). With nearly all mutations seen in LQTS patients identified as loss-of-function and 50% of those loss-of-function mutations identified in the KCNQ1 subunit (*Hedley et al., 2009*; *Ackerman et al., 2011*), enhancing and activating $I_{Ks}$ currents has long been suggested as a promising therapeutic approach for treating LQTS. It is a curiosity of $I_{Ks}$ channels that the known activators fall into two general groupings: those that work best on the α-subunit, KCNQ1, alone, including zinc pyrithione, L-364,373, and one of the most studied activators, ML277 (*Mattmann et al., 2012*; *Yu et al., 2013*; *Xu et al., 2015*; *Eldstrom et al., 2021*); and those that work best in the presence of the auxiliary β-subunit, KCNE1, compounds like mefenamic acid and DIDS (4,4'-diisothiocyano-2,2'- stilbene disulfonic acid) (*Abitbol et al., 1999*; *Wang et al., 2020*).

Using cryo-EM, we recently visualized the binding of ML277 deep in the central core of KCNQ1 channels in a pocket lined inferiorly by the S4-S5 linker, laterally by the S5 and S6 helices of two separate subunits, and above by pore domain residues (*Willegems et al., 2022*). The location of the binding pocket and its structural inter-relationships help to explain the underlying mechanism of action of ML277, its specificity for KCNQ1, and the lack of efficacy due to steric hindrance in the presence of a β-subunit. However, it is known neither where activators of $I_{Ks}$ bind that *require* the presence of the β-subunit, such as phenylboronic acid (*Mruk and Kobertz, 2009*), hexachlorophene, stilbenes such as DIDS and SITS (4-acetamido-4'-isothiocyanatostilbene-2,2'-disulfonic acid), and diclofenac acid derivatives such as mefenamic acid (*Abitbol et al., 1999*; *Zheng et al., 2012*; *Wang et al., 2020*), nor how they mediate their activator action. We have some clues for DIDS and mefenamic acid that their binding sites are not in the central channel core, as is the case for ML277, but depend on KCNE1 β-subunit residues at the extracellular surface of the channel, residues 39–43 in KCNE1 for DIDS (*Abitbol et al., 1999*) and K41 for mefenamic acid (*Wang et al., 2020*). Our recent cysteine scanning data revealed that although other extracellular KCNE1 residues in the same region to varying degrees impacted the effect of mefenamic acid, only the K41C mutation completely abolished mefenamic acid effect up to a concentration of 1 mM (*Wang et al., 2020*). Previous cross-linking studies have identified key interactions between this extracellular region of KCNE1 and the S1 and S6 transmembrane segments of KCNQ1 (*Xu et al., 2008*; *Chung et al., 2009*), suggesting that residues in either of these regions could also provide important clues to explain mefenamic acid's mechanism of action.

To further explore the dependence of residues in KCNE1 and those in adjacent KCNQ1 sites on the binding of mefenamic acid to $I_{Ks}$, we first examined the role of K41C in preventing the drug effect. Docking in combination with molecular dynamics (MD) simulations of mefenamic acid binding to $I_{Ks}$ followed by mutagenesis were used to map out critical KCNE1 and KCNQ1 residues. Further, we expanded on the idea that the stilbene, DIDS (*Abitbol et al., 1999*), which is structurally quite different from mefenamic acid, shares a common binding site. Our results showed that both compounds bind in the same general region formed by elements of the pore and S6 domains of KCNQ1 and the near extracellular region of KCNE1, but depend on different critical residues for their binding stability. Exposure of the channel complex to either compound induces subtle structural changes that subsequently stabilize the conformation of the S1/outer pore/S6 of KCNQ1 and slow the $I_{Ks}$ deactivation gating kinetics. The results suggest the existence of a common drug-induced binding site and a mechanism of action for small molecule $I_{Ks}$ activators that is distinct from that of specific compounds that activate KCNQ1 alone.

## Results

### The mefenamic acid binding site on the KCNQ1/KCNE1 complex

Exposure of wild type $I_{Ks}$ complexes (4:4 ratio; WT EQ) to 100 μM mefenamic acid transforms the slowly activating $I_{Ks}$ current into one with an almost linear waveform and completely inhibits tail current decay at –40 mV (*Figure 1A*; *Abitbol et al., 1999*; *Unsöld et al., 2000*; *Wang et al., 2020*). G-V relations obtained from peak initial tail currents show that 100 μM mefenamic acid hyperpolarizes the G-V ($\Delta V_{1/2}$ = -105.7 mV, *Figure 1B and C*) and decreases the slope (control $k$=19.4 mV, Mef $k$=41.3 mV,

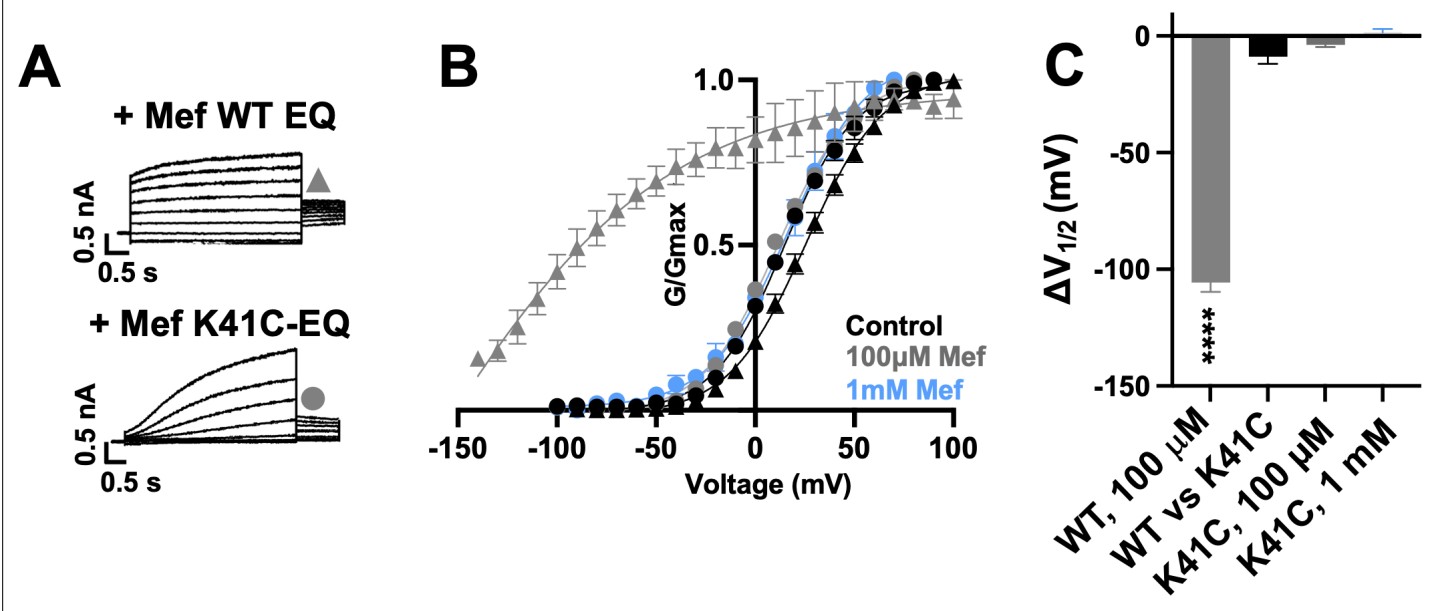

**Figure 1.** K41C-KCNE1 mutants prevent the agonist effect of mefenamic acid. (**A**) Current traces of WT EQ (top) and K41C-EQ (bottom) in the presence of 100 µM mefenamic acid (Mef). A 4 s protocol was used with pulses from –150 mV or higher to +100 mV, in 10 mV steps, followed by a repolarization step to –40 mV for 1 s. Holding potential and interpulse interval were –80 mV and 15 s, respectively. (**B**) G-V plots obtained from WT EQ tail currents (triangles) and K41C-EQ (circles) in the absence (control: black) and presence of Mef (100 µM: grey; 1 mM: light blue). Boltzmann fits were: WT EQ control (n=6): $V_{1/2}$ = 25.4 mV, $k$=19.4 mV; WT EQ 100 µM Mef (n=3): $V_{1/2}$ = -80.3 mV, $k$=41.3 mV; K41C-EQ control (n=4): $V_{1/2}$ = 15.2 mV, $k$=18.4 mV; K41C-EQ 100 µM Mef (n=4): $V_{1/2}$ = 11.4 mV, $k$=19.4 mV; and K41C-EQ 1 mM Mef (n=3): $V_{1/2}$ = 16.7 mV, $k$=19.8 mV. Error bars shown are SEM. (**C**) Summary plot of $V_{1/2}$ change ($\Delta V_{1/2}$) for WT EQ in the presence of 100 µM mefenamic acid, WT EQ vs K41C-EQ in control and K41C-EQ in the presence of 100 µM and 1 mM mefenamic acid. Data are shown as mean ± SEM and unpaired t-test was used. **** denotes a significant $\Delta V_{1/2}$ compared to control where p<0.0001.

*Table 1*). We previously showed that introduction of a cysteine mutation at residue K41 in all four KCNE1 subunits (4:4 ratio of mutant K41C-KCNE1 to KCNQ1; K41C-EQ) itself had only a minor effect on the G-V, but prevented changes to currents and the G-V relationship on exposure to 100 µM or 1 mM mefenamic acid (*Figure 1B and C* and *Table 1*; *Wang et al., 2020*). The data suggest that residue K41 in KCNE1 is critical to the action of mefenamic acid, but give no information why K41 is so important, and prompted us to question how this residue and other adjacent residues in KCNE1, and those nearby on KCNQ1 acted together to form a binding pocket for mefenamic acid.

## Mefenamic acid binding site predicted by molecular modeling

Initially, to visualize potential drug binding sites and understand how mutation of KCNE1 and KCNQ1 residues might prevent drug action, in-silico experiments of drug docking with subsequent MD simulations were performed on a model of $I_{Ks}$ channels. The $I_{Ks}$ model was constructed based on the recent cryo-EM structure of KCNQ1-KCNE3, which is thought to represent the activated-open state of the channel complex (*Sun and MacKinnon, 2020*). Taking into consideration the sequence similarity of KCNE1 and KCNE3 subunits in their transmembrane segments, it has been suggested that the main interface of these subunits with KCNQ1 is preserved in this region. Our initial data indicated that the extracellular residues of KCNE1 are involved in the action of $I_{Ks}$ activators, so we constructed a model where external KCNE3 residues R53-Y58 were substituted with homologous KCNE1 residues, D39-A44 (*Figure 2A*). The resulting 4:4 $I_{Ks}$ channel complex was termed pseudo-KCNE1-KCNQ1, ps-$I_{Ks}$, and *Figure 2B* shows the essential elements of the ps-$I_{Ks}$ subunits which form the extracellular interface of KCNQ1 and KCNE and served as a basis for drug docking and MD simulations. Details of the docking procedure are described in the Materials and methods section and schematically summarized in *Figure 2—figure supplement 1*.

Briefly, conformational sampling was performed on ps-$I_{Ks}$ residues D39-A44, and conformations showing the lowest free energy were selected for docking using a four-dimensional (4D) docking approach to find the best binding pose of the ligand. A conformation with the best docking score

**Table 1.** Mean $V_{1/2}$ of activation (mV) and slope values (k-factor, mV) in the absence and presence of mefenamic acid for fully saturated $I_{Ks}$ channel complexes.

A statistical difference in $V_{1/2}$ compared to control is shown as p-value, determined using an unpaired t-test. NS denotes not significant. Values are shown ± SEM.

| | Control | | | 100 µM or 1 mM mefenamic Acid* | | | |
|---|---|---|---|---|---|---|---|
| | $V_{1/2}$ | k-factor | n | $V_{1/2}$ | k-factor | n | p-value |
| WT EQ | 25.4±2.4 | 19.4±1.2 | 6 | −80.3±4.1 | 41.3±8.4 | 3 | <0.0001 |
| | | | | 11.4±1.0 | 19.4±0.8 | 4 | <0.05 |
| K41C-EQ | 15.2±1.1 | 18.4±1.7 | 4 | 16.7±2.0 | 19.8±1.4 | 3 | NS |
| L42C-EQ | 68.9±1.5 | 21.5±3.7 | 3 | 31.8±0.4 | 14.8±4.27 | 3 | <0.01 |
| E43C-EQ | 18.3±10.0 | 25.7±1.5 | 6 | 14.4±5.6 | 22.8±1.3 | 6 | NS |
| A44C-EQ | 4.1±1.8 | 17.6±1.4 | 4 | −5.6±2.7 | 18.3±2.0 | 4 | <0.05 |
| Y46A-EQ† | 76.4±1.5 | 57.3±1.6 | 3 | −29.8±4.1 | 18.9±3.1 | 3 | <0.05 |
| EQ-W323A | 47.8±2.7 | 23.7±2.2 | 4 | 33.7±1.2 | 28.5±3.1 | 3 | <0.05 |
| EQ-W323C | 54.0±2.1 | 22.2±2.1 | 4 | 27.5±3.3 | 25.6±1.8 | 4 | <0.05 |
| EQ-V324A | 34.4±2.3 | 20.7±1.1 | 6 | 15.5±2.3 | 27.9±1.6 | 5 | <0.05 |
| EQ-V324W | 41.0±2.4 | 18.4±0.3 | 4 | 27.3±6.0 | 25.5±1.3 | 4 | NS |
| EQ-Q147C | 63.9±5.8 | 25.4±2.0 | 4 | 26.3±5.7 | 32.7±1.7 | 4 | <0.05 |
| EQ-Y148C | 36.8±0.3 | 20.3±0.4 | 4 | 17.5±4.0 | 36.7±3.7 | 4 | <0.05 |

*For K41C-EQ, the concentration of mefenamic acid used was either 100 µM (upper row values) or 1 mM (lower row values). For all other constructs, 100 µM mefenamic acid was used.

†An estimation of the activation $V_{1/2}$ was calculated from a right shifted, non-saturating GV curve.

(*Figure 2B*) shows mefenamic acid binding to the pocket formed between extracellular KCNE1 residues, the external S6 transmembrane helix of one subunit, and the S1 transmembrane domain of the neighboring subunit and the pore turret of a third subunit. The estimated volume of the pocket in the mefenamic acid bound state is ~307 Å$^3$ with a hydrophobicity value of ~0.65 kcal/mol (*Figure 2D*). The free energy of the mefenamic acid-ps-$I_{Ks}$ interaction estimated by the MM/GBSA method from analysis of 300 ns MD simulations was −37.9±1.22 kcal/mol (Figure 4A, Table 4), while a similar value of −31.8±1.46 kcal/mol was calculated using the Poisson-Boltzmann surface area (MM/PBSA) model. The MM/GBSA and MM/PBSA data were used to identify ps-$I_{Ks}$ residues contributing to the free energy of interaction with mefenamic acid for further analysis (*Figure 2—figure supplement 2*), although it should be noted that root mean square deviation (RMSD) analysis of trajectories indicated that the binding was dynamic as there were other binding conformations possible (*Figure 2—figure supplement 3*). Mefenamic acid in this complex was stabilized by two hydrogen bonds formed between the drug and ps-KCNE1 residues Y46 and E43 (*Figure 2C*). Analysis of MD simulations showed that these two H-bonds were the most stable, the Y46 H-bond to mefenamic acid had a mean frequency of 79.7%, and E43 22.5% of the drug bound time (*Figure 2—figure supplement 4*; *Table 2*).

The energy decomposition per amino acid using MM/GBSA and MM/PBSA methods revealed several residues in KCNQ1 and ps-KCNE1 with significant contributions to mefenamic acid coordination (*Figure 2C*, *Figure 2—figure supplement 2*). As expected, these are the KCNE1 residues located at the external region of the auxiliary subunit – amino acids K41 to Y46. In addition, residues W323 and V324 located on the S6 helix as well as residues L142, Q147, and Y148 located on the S1 helix exhibited the lowest interaction free energy. We focused on functional validation of the KCNE1 residues K41, L42, E43, A44, and Y46, and KCNQ1 residues W323, V324, L142, Q147, and Y148 by mutation to cysteine, alanine and/or tryptophan, and examining the sensitivity of fully saturated EQ channel complexes to 100 µM mefenamic acid. Although A300, located in the turret region was identified as potentially important (but less so than KCNE residues and W323), we could not get adequate expression from A300C for functional analysis.

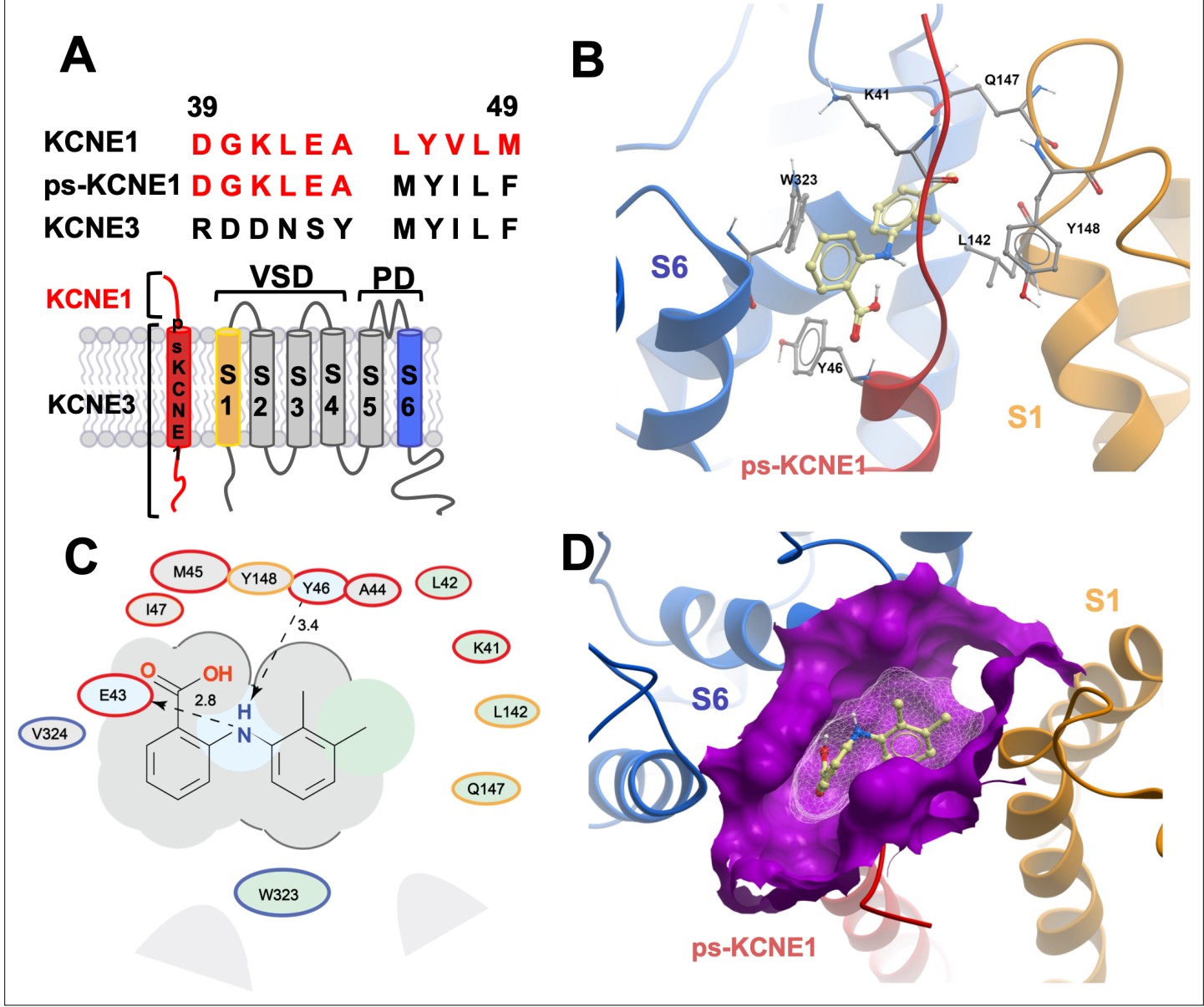

**Figure 2.** MD prediction of mefenamic acid binding site in the ps-$I_{Ks}$ model. (**A**) Pseudo-KCNE1 (ps-KCNE1) used to predict Mef binding site. Extracellular residues of KCNE1 (top), ps-KCNE1 (middle) and KCNE3 (bottom). Below, cartoon topology of the single transmembrane ps-KCNE1 β-subunit and the six transmembrane KCNQ1 α-subunit. S1-S4 transmembrane segments form the voltage sensor domain and S5-S6 form the pore domain. (**B**) Binding pose of Mef (yellow) in the external region of the ps-$I_{Ks}$ channel complex obtained with docking (side view). Pore domain residues are blue, ps-KCNE1 subunit red, and the VSD of a neighbouring subunit is in yellow. (**C**) Ligand interaction map of Mef with ps-$I_{Ks}$ from molecular docking. Size of residue ellipse is proportional to the strength of the contact. The distance between the residue label and ligand represents proximity. Grey parabolas represent accessible surface for large areas. Light grey ellipses indicate residues in van der Waals contacts, light green ellipses are hydrophobic contacts, and light blue are H-bond acceptors. Red borders indicate KCNE1, yellow are KCNQ1 VSD, and blue are pore residues. Dashed lines indicate H-bonds. The 2D diagram was generated by ICM pro software with cut-off values for hydrophobic contacts of 4.5 Å and hydrogen bond strength of 0.8. Further details in Materials and methods. (**D**) Mef binding pose observed in MD simulations in space-fill to highlight pocket formed by external S1 (yellow), S6 (blue) transmembrane domains of KCNQ1, and extracellular region of the ps-KCNE1 subunit (red). This binding conformation is the most frequent binding pose of Mef observed in ~50% of frames and corresponds to the blue-framed conformation in *Figure 2—figure supplement 3*.

The online version of this article includes the following figure supplement(s) for figure 2:

**Figure supplement 1.** Drug docking and MD simulation workflow.

**Figure supplement 2.** Energy decomposition per amino acid for mefenamic acid binding to ps-$I_{Ks}$.

*Figure 2 continued on next page*

*Figure 2 continued*

**Figure supplement 3.** Clustering of MD trajectories based on mefenamic acid conformations.

**Figure supplement 4.** H-bonds formed between mefenamic acid and protein residues in its binding site during MD simulations.

## Mutational impact on EQ current changes induced by mefenamic acid

The effect of mutations on the current waveform and tail current response to 100 µM mefenamic acid treatment was examined on $I_{Ks}$ channels identified as: x-EQ-y where 'x' denotes a KCNE1 mutation and 'y' denotes a KCNQ1 mutation. In the absence of mefenamic acid (control), most mutations, with the exception of EQ-L142C (*Figure 3—figure supplement 2*), produced slowly activating currents with rapid tail current decay (*Figure 3A*).

In the presence of 100 µM mefenamic acid, the waveforms of WT EQ, S6 and pore mutations EQ-V324A, EQ-V324W, EQ-Q147C and EQ-Y148C were all transformed into ones with instantaneous current onset and slowed tail current decay (*Figure 3A*, *Figure 3—figure supplement 1*). Only the EQ-W323A waveform and tail current were largely unaffected by 100 µM mefenamic acid (like K41C-EQ, *Figure 1*). The EQ-W323C and A44C-EQ current waveforms were also unchanged by 100 µM mefenamic acid, but their tail current decay was slowed (*Figure 3A*, *Figure 3—figure supplement 1*). We interpreted this to mean that mefenamic acid binds to EQ-W323C and A44C-EQ mutant open channels and slows closing, but that the drug-channel complex is less stable and mefenamic acid unbinds during the interpulse interval, which relieves drug action between pulses. The summary of the normalized response of the different mutants to mefenamic acid is shown in *Figure 3D*. G-V plots were obtained from the tail current amplitudes in the absence and presence of 100 µM mefenamic acid, and unlike WT EQ (*Figure 1*), minimal change in the shape and position of the EQ-W323A G-V plot was seen after exposure to 100 µM mefenamic acid $V_{1/2}$ shift of –14.1 mV compared with –105.7 mV seen in WT EQ (*Figure 1*, *Figure 3B and E*, and *Table 1*). For the W323C mutation, a less drastic decrease in size compared to alanine, the $V_{1/2}$ shift seen with 100 µM mefenamic acid increased (EQ-W323C $\Delta V_{1/2}$ = -26.5 mV), suggesting that the size of the W323 residue is important (*Figure 3E* and *Table 1*). However, when the neighboring V324 KCNQ1 residue was mutated to a smaller (alanine) or bulkier (tryptophan) residue they both showed the same response to mefenamic acid. Both the V324W and V324A mutations reduced $V_{1/2}$ shifts caused by 100 µM mefenamic acid to between –14 to –19 mV (*Figure 3E*, *Table 1*, *Figure 3—figure supplement 1*).

Extracellular S1 residues identified in the MD simulations also proved to be important for mefenamic acid binding, although less so than W323 and K41 residues found in the KCNQ1 extracellular regions

**Table 2.** Mean frequency of H-bonds between MEF or DIDS and residues in ps- $I_{Ks}$ during 500 ns MD simulations or until drug unbound.

MEF357-N1 indicates the nitrogen of aminobenzoic acid; MEF357:O1 and O2 are the oxygens of the aminobenzoic acid; DDS357:O1-O6 are the oxygens of the sulfonic acids; DDS357:N1 and N2 are nitrogens of the isothiocyanates. n=5 for each.

|      | Donor    | Hydrogen   | Acceptor  | % Frames with H-Bonds | SEM  |
|------|----------|------------|-----------|------------------------|------|
| MEF  | MEF-N1   | MEF-H1     | GLU43-O   | 22.5                   | 10.2 |
|      | TYR46-N  | TYR46-HN   | MEF-O2    | 79.7                   | 4.2  |
|      | ILE47-N  | ILE47-HN   | MEF-O2    | 7.9                    | 2.7  |
|      | DIDS-O2  | DDS357-H10 | TRP323-NE1| 12.9                   | 2.1  |
|      | DIDS-O1  | DDS357-H9  | GLU43-OE2 | 25.1                   | 4.0  |
|      | DIDS-O1  | DDS357-H9  | GLU43-OE1 | 25.2                   | 4.6  |
|      | ILE47N   | ILE47-HN   | DIDS-O5   | 22.0                   | 3.2  |
|      | TYR46-N  | TYR46-HN   | DIDS-O6   | 56.8                   | 10.4 |
|      | TYR46-N  | TYR46-HN   | DIDS-O5   | 7.1                    | 1.8  |
|      | TYR299N  | TYR299-HN  | DIDS-N2   | 11.9                   | 0.6  |
| DIDS | SER298-OG| SER298-HG1 | DIDS-N2   | 26.0                   | 10.6 |

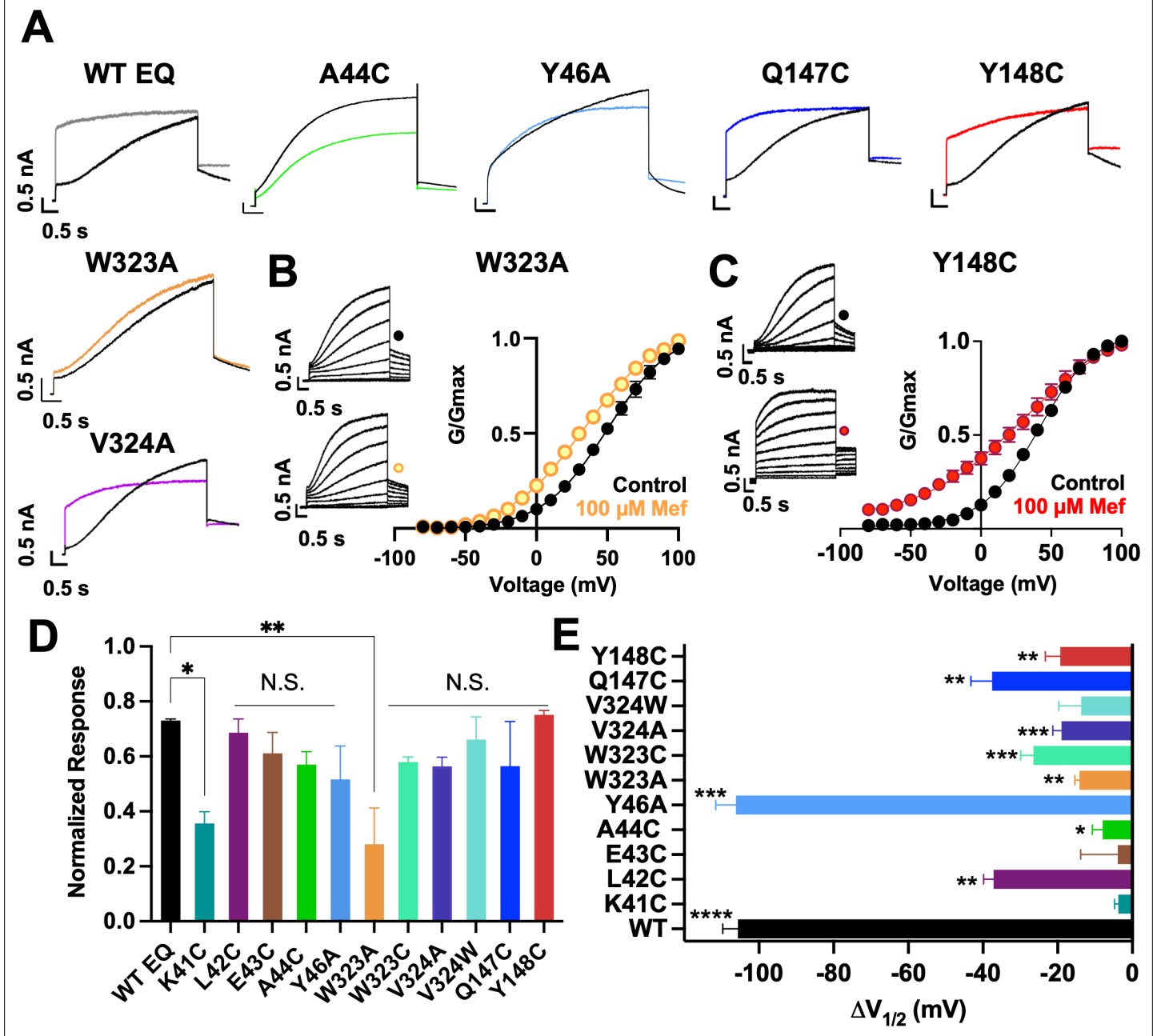

**Figure 3.** Current waveform and G-V changes induced by mefenamic acid in binding site mutants. (**A**) Current traces from WT EQ and key residue mutants in control (black) and 100 µM Mef (colors). (**B**) EQ-W323A and (**C**) EQ-Y148C current traces in control (top) and presence of 100 µM Mef (below). G-V plots in control (black) and presence of 100 µM Mef (colors). Boltzmann fits were: EQ-W323A control (n=4): $V_{1/2}$ = 47.8 mV, $k$=23.7 mV; EQ-W323A Mef (n=3): $V_{1/2}$ = 33.7 mV, $k$=28.5 mV; EQ-Y148C control (n=4): $V_{1/2}$ = 36.8 mV, $k$=20.3 mV; and EQ-Y148C Mef (n=4): $V_{1/2}$ = 17.5 mV, $k$=36.7 mV. Voltage steps were from –80 mV to +100 mV for 4 s, followed by a 1 s repolarization to –40 mV. Interpulse interval was 15 s. Error bars shown are ± SEM. (**D**) Summary plot of the normalized response to 100 µM Mef (see Materials and methods). Data are shown as mean + SEM and one-way ANOVA statistical test was used. \*\*p<0.01 and \*p<0.05 denote a significantly reduced response compared to WT EQ. N.S. denotes not significant. (**E**) Change in $V_{1/2}$ ($\Delta V_{1/2}$) for WT EQ and each $I_{Ks}$ mutant in control versus mefenamic acid. Data are shown as mean -SEM and unpaired t-test was used, where \*p<0.05, \*\*p<0.01, \*\*\*p<0.001, \*\*\*\*p<0.0001 indicate a significant change in $V_{1/2}$ comparing control to the presence of the drug. n-values for mutants in D and E are stated in *Table 1*.

The online version of this article includes the following figure supplement(s) for figure 3:

**Figure supplement 1.** Current waveform and G-V changes induced by mefenamic acid for all binding site mutants.

**Figure supplement 2.** Augmented activation of EQ-L142C in the absence of mefenamic acid.

**Table 3.** Mean $V_{1/2}$ of activation (mV) and slope values (*k*-factor, mV) for WT EQ treated with 100 μM mefenamic acid, and untreated mutant EQ-L142C.

Interpulse interval used is as indicated. Values are shown ± SEM. An interpulse interval of 7 s created such a dramatic change in the shape of the EQ-L142C G-V plot (see *Figure 3—figure supplement 2*) that a Boltzmann curve could not be fit and $V_{1/2}$ and *k*-values are therefore not available.

|  | $V_{1/2}$ | *k-factor* | n |
|---|---|---|---|
| WT EQ +100 μM mefenamic acid: Interpulse interval 15 s | -80.3±4.1 | 41.3±8.4 | 3 |
| WT EQ +100 μM mefenamic acid: Interpulse interval 30 s | 26.7±10.5 | 66.6±28 | 8 |
| EQ-L142C control: Interpulse interval 15 s | -80.3±4.5 | 30.0±2.9 | 6 |
| EQ-L142C control: Interpulse interval 30 s | -28.7±19 | 62.5±9.6 | 3 |

of the S6 segment and KCNE1, respectively. Compared to WT, EQ-Y148C reduced the $V_{1/2}$ shift after exposure to 100 μM mefenamic acid (EQ-Y148C $\Delta V_{1/2}$ = -19.3 mV) and lessened the slope of the G-V relationship (*Figure 3C*). The Q147C mutant on the other hand, only partially prevented the $V_{1/2}$ shift observed after mefenamic acid treatment (EQ-Q147C $\Delta V_{1/2}$ = -37.6 mV, *Figure 3E*).

Other KCNE1 residues located at the N-terminal limit of the KCNE1 transmembrane segment (L42, E43, A44, Y46) were also investigated. Unlike with K41C, but similar to WT EQ, tail current decay was inhibited in the presence of 100 μM mefenamic acid in all of these mutants, reflected by the normalized response (*Figure 3D*, *Figure 3—figure supplement 1*). In addition, a reduced but still significant shift in $V_{1/2}$ was observed with 100 μM mefenamic acid compared to control in most mutants tested, except K41C. (K41C $\Delta V_{1/2}$ = -3.8 mV; L42C $\Delta V_{1/2}$ = -37.1 mV; E43C $\Delta V_{1/2}$ = -3.9 mV; A44C $\Delta V_{1/2}$ = -9.7 mV; *Figure 3E*). Representative current traces and G-V curves for all mutants are shown in *Figure 3— figure supplement 1*. Consistent with previous literature (*Gofman et al., 2012*; *Wang et al., 2012*; *Kuenze et al., 2020*), Y46C-EQ in control conditions produced a current with faster activation and a complex GV curve which made analysis and an assignment of slope and $V_{1/2}$ difficult, though a small left shift in the GV curve was visible (data not shown). In lieu of Y46C, G-V data from the Y46A-EQ mutant showed a potent effect of mefenamic acid, equivalent to that of WT with a $\Delta V_{1/2}$ = -106.2 mV (*Figure 3—figure supplement 1D*). These results suggest that mutation of residues further away from the N-terminus of KCNE1 than K41 has diminishing effects on the activator action of mefenamic acid.

## Augmented activation of EQ-L142C in the absence of mefenamic acid

Unlike other mutations, the control EQ-L142C current waveform displayed an almost instantaneous current onset, and tail currents showed no decay with our standard protocol (*Figure 3—figure supplement 2A*). As the control EQ-L142C current waveform qualitatively resembled WT EQ currents in the

**Table 4.** Average free interaction energies of MEF-bound ps-$I_{Ks}$ complexes calculated according to MM/GBSA methods from three independent MD simulation runs using the AMBER force field.

Mean values are in kcal/mol, ± SD. For W323A and K41C mutations, calculations correspond to interval of simulations before the detachment of ligand from the molecular complex. Note that unbinding occurred in K41C and W323A in all 3 simulations with 300 ns duration.

| Run | Method | ps-$I_{Ks}$ | W323A | Y46C | K41C |
|---|---|---|---|---|---|
|  | GBSA | -39.3±4.1 | -13.0±5.1 | -31.2±2.9 | -10.5±3.8 |
| I |  | - | Unbinding after ~75 ns |  | Unbinding after ~25 ns |
|  | GBSA | -39.0±3.3 | -23.1±3.1 | -32.2±3.8 | -17.0±2.6 |
| II |  | - | Unbinding after ~120 ns | - | Unbinding after ~70 ns |
|  | GBSA | -35.5±2.6 | -22.6±4.1 | -28.8±5.7 | -17.2±2.5 |
| III |  | - | Unbinding after ~85 ns | - | Unbinding after ~20 ns |

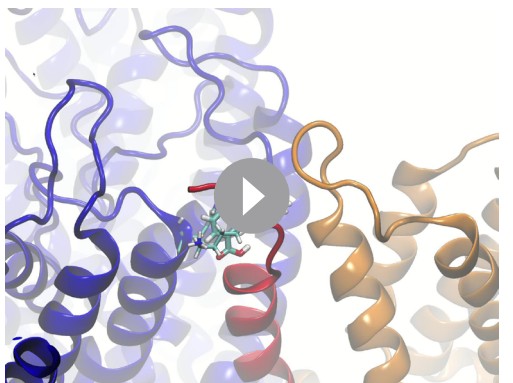

**Video 1.** MD simulations at the molecular level of binding of mefenamic acid to ps-$I_{Ks}$, and K41C-, W323A-, Y46C- ps-$I_{Ks}$ mutants. Note that videos may be shorter than the actual 500 ns simulations if drugs do not remain bound. Mefenamic acid binding to ps-$I_{Ks}$. W323, and K41 side chains are shown.

https://elifesciences.org/articles/87038/figures#video1

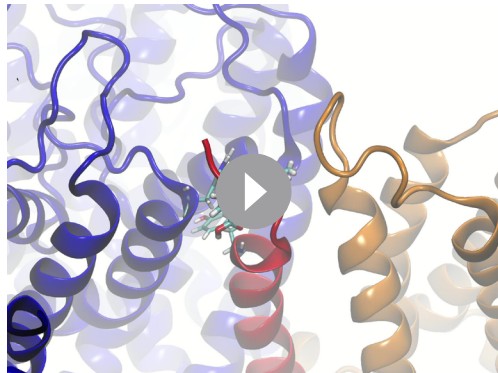

**Video 2.** Mefenamic acid binding to K41C ps-$I_{Ks}$. K41C, W323, and Y46 side chains are shown.

https://elifesciences.org/articles/87038/figures#video2

presence of 100 μM mefenamic acid, G-V plots at different interpulse intervals were compared. At the standard interval of 15 s, the EQ-L142C G-V plot (blue) closely overlapped that of WT EQ in the presence of Mef (solid grey), and at a 30 s interpulse interval, the position of the WT EQ +Mef (grey open circles) and EQ-L142C plots (green) both depolarized significantly (*Figure 3—figure supplement 2B*). It appeared that the L142C mutation augmented channel activation as much as 100 μM mefenamic acid on WT channels. In addition, mefenamic acid made the EQ-L142C G-V plot voltage independent, overlapping with the G-V obtained for EQ-L142C with a 7 s interpulse interval (pink and red respectively, *Figure 3—figure supplement 2B*). The data indicate that EQ-L142C is still responsive to mefenamic acid but does so from a heightened state of activation (*Table 3*).

## MD simulations of mutant ps-$I_{Ks}$ channels exposed to mefenamic acid

To explore the role played by critical residues, in-silico homologous mutations K41C, Y46C and W323A were introduced into the mefenamic acid-bound ps-$I_{Ks}$ channel (Mef-ps-$I_{Ks}$) and the stability of the Mef-mutant complexes was assessed in MD simulations. Remarkably, mefenamic acid detached from mutant K41C and W323A channels within 120 ns in all three independent MD simulations with an AMBER force field (*Table 4*). In contrast, mefenamic acid remained bound during the entire simulation time to WT and Y46C-ps-$I_{Ks}$ complexes (*Table 4*). The last mutant was tested as we could not determine the functional effect of mefenamic acid on this residue in electrophysiological experiments, but in the simulation, Y46C-ps-$I_{Ks}$ behaved like Y46A from electrophysiological data (*Figure 3*, *Figure 3—figure supplement 1*). Mefenamic acid unbound from A44C-ps-$I_{Ks}$ channel complexes within 80, 100, and 110 ns during three different simulations, changing binding pose several times before doing so.

Similar results were obtained in MD simulations with the CHARMM force field where Mef-ps-$I_{Ks}$ was embedded in the lipid membrane (*Video 1*), although in two of five 1250 ns long-duration simulations we observed unbinding of Mef from WT ps-$I_{Ks}$ complexes, which suggested that the lipid environment allowed a more dynamic interaction between Mef and $I_{Ks}$. In contrast, Mef left the binding site in all five 500 ns runs for W323A and K41C mutants (*Figure 4—figure supplement 1*, *Videos 2 and 3*), and when the Y46C mutant was placed in a lipid environment, mefenamic acid left the binding site or changed its binding

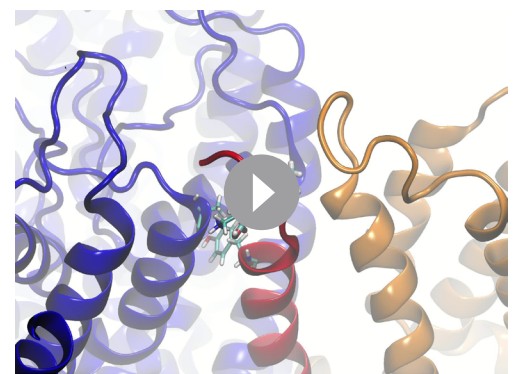

**Video 3.** Mefenamic acid binding to W323A ps-$I_{Ks}$. K41, W323A, and Y46 side chains are shown.

https://elifesciences.org/articles/87038/figures#video3

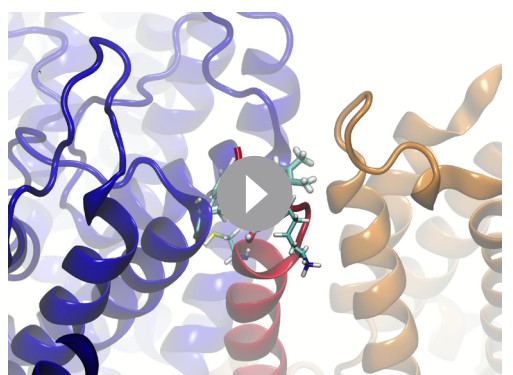

**Video 4.** Mefenamic acid binding to Y46C ps-$I_{Ks}$. K41, W323, and Y46C side chains are shown.

https://elifesciences.org/articles/87038/figures#video4

conformation in three of five 500 ns MD simulations (*Video 4*). A significantly *reduced* free interaction energy (ΔG) of ligand binding for K41C and W323A mutant Mef-ps-$I_{Ks}$ complexes in 300 ns simulations was observed compared to WT Mef-ps-$I_{Ks}$ (*Figure 4A*, *Table 4*), and the small but statistically significant change in free energy observed for the Y46C mutant complex confirms that the Y46 residue is not as important as K41 and W323 for mefenamic acid binding. It should be noted that the absolute values of ΔG or corresponding dissociation constants ($K_d$) calculated from MD simulations do not reflect apparent ΔG and $K_d$ values determined from electrophysiological experiments or biochemical essays.

The flexibility of the external ps-KCNE1 protein residues of the mutants W323A, K41C, and Y46C ps-$I_{Ks}$ channels was also analyzed from 300 ns duration trajectories, by monitoring their average root mean square fluctuation (RMSF) during the last 100 ns of simulations after the detachment of mefenamic acid from the molecular complex or in the case of Y46C without introduction of the ligand. The RMSF values obtained from the mutant channels were then compared to that of WT ps-$I_{Ks}$ channels where MD simulations of the same duration were conducted after removing the mefenamic acid molecule from the complex (*Figure 4B*). The results indicate that K41C and W323A mutations modestly increased the RMSF of D39-A44 ps-KCNE1 residues when compared to the WT ps-$I_{Ks}$ channel complex without mefenamic acid bound (*Figure 4C and D*). The Y46C mutation seems to impact the mobility of the N-terminal portion of KCNE1 much less (*Figure 4E*), perhaps owing to its location further into the membrane. These results suggest that the side chains of K41 and W323 residues normally stabilize the conformation of the external region of KCNE1, so that mutation of these residues increases random motion and reduces the ability of drugs like mefenamic acid to remain bound at this site.

## Mutational impact on $I_{Ks}$ current changes induced by DIDS

To establish whether the binding pocket for mefenamic acid can be generalized to other $I_{Ks}$ activators, we also examined the binding of the structurally unrelated $I_{Ks}$ activator, DIDS. Stilbenes such as DIDS (*Figure 5A*) also activate $I_{Ks}$ (*Abitbol et al., 1999*; *Bollmann et al., 2020*), and given their molecular differences from fenamates, it is of interest to explore common structural and dynamic features of their binding to the $I_{Ks}$ channel complex. 100 µM DIDS had no effect on endogenous currents in GFP-transfected tsA201 cells but treatment of WT EQ channels with 100 µM DIDS transformed the slowly activating waveform into one with faster onset (although not instantaneous like mefenamic acid) and inhibited tail current decay (*Figure 5B*). G-V plots were obtained from the tail amplitudes in the absence and presence of 100 µM DIDS (*Figure 5C and D*). The overall shape, slope, and $V_{1/2}$ of the WT EQ G-V relationship changed with 100 µM DIDS in the direction of increased activation ($\Delta V_{1/2}$ = -46.6 mV; *Table 5*; control $k$=20.3 mV, DIDS $k$=25.3 mV). However, the effects on WT EQ were less pronounced with DIDS than mefenamic acid at the 100 µM concentration.

The ps-$I_{Ks}$ construct and docking procedures used previously to explore the binding site of mefenamic acid were used as a basis for DIDS docking and subsequent MD simulations. We found that DIDS also bound to a location formed between extracellular KCNE1 residues, the external S6 transmembrane helix of one subunit, the S1 transmembrane domain of the neighboring subunit, and the pore turret of the opposite subunit (*Figure 6A*). In docking experiments, DIDS is stabilized by its hydrophobic and van der Waals contacts with KCNQ1 and KCNE1 subunits as well as by two hydrogen bonds formed between the drug and ps-KCNE1 residue L42 and KCNQ1 residue Q147 (*Figure 6B*). These hydrogen bonds however were not stable during MD simulations and instead H-bonding occurs between E43, Y46, and I47 in KCNE1 as well as W323, S298 and Y299 in KCNQ1 (*Figure 6—figure supplement 1*, *Table 2*). The energy decomposition per amino acid using MM/GBSA and MM/PBSA methods based on 300 ns simulations revealed several residues in KCNQ1 and

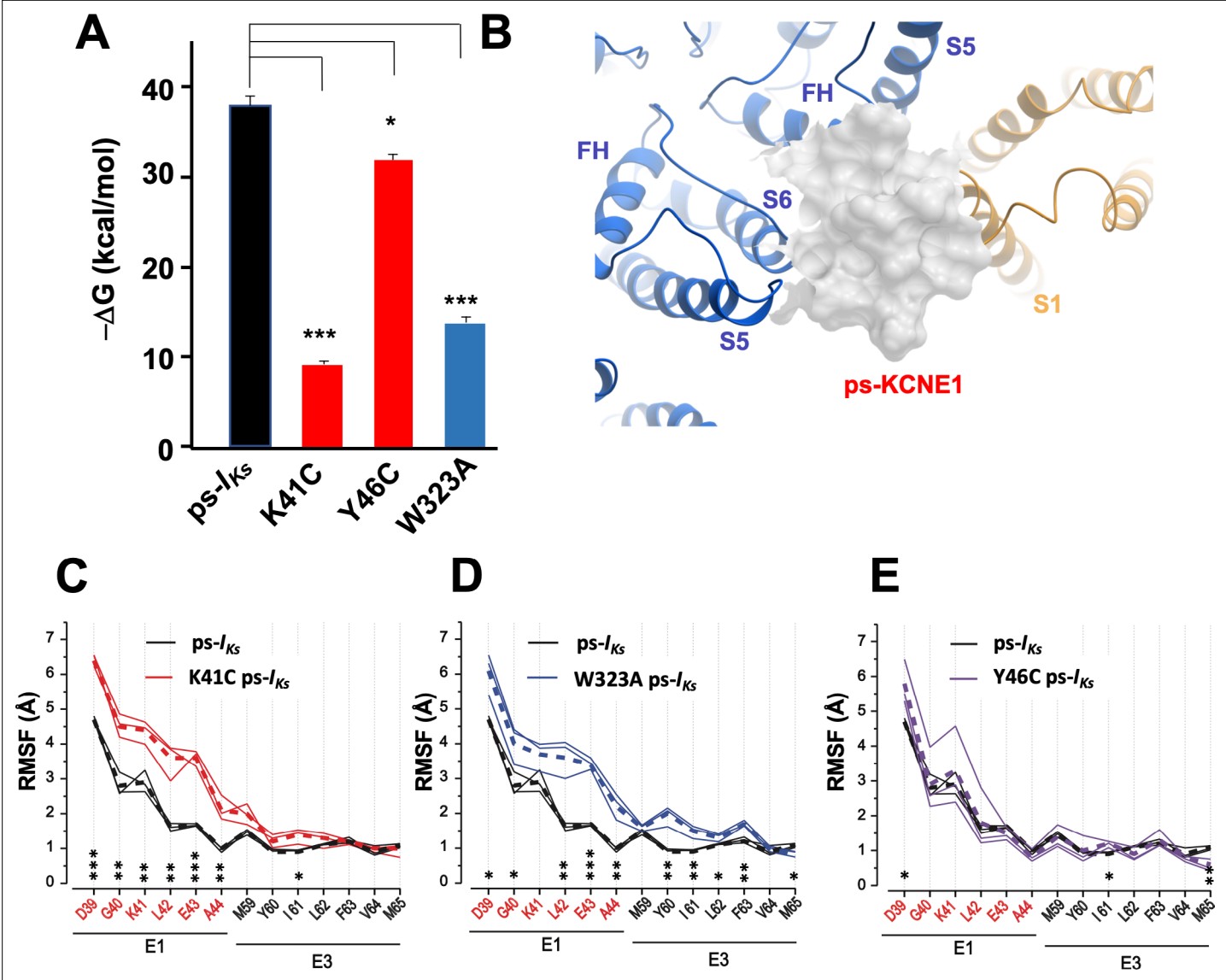

**Figure 4.** K41C, Y46C, and W323A mutant impact on mefenamic acid binding energy and flexibility of external KCNE1 residues. (**A**) Average free interaction energy of Mef-bound ps-$I_{Ks}$ complexes calculated using MM/GBSA methods from three independent MD simulation runs. For K41C and W323A mutations, calculations correspond to interval of simulations before the detachment of ligand from the molecular complex. * and *** denote significant differences in average free interaction energy compared to ps-$I_{Ks}$. Student`s unpaired t-test was used for comparison of groups. (**B**) Surface representation of ps-$I_{Ks}$ after removal of Mef. The pore residues are in blue and the VSD of a neighbouring subunit is yellow. (**C–E**) Root mean square fluctuations (RMSF) of ps-KCNE residues (Å) in the ps-$I_{Ks}$ complex during the last 100 ns of simulations. Three separate MD simulations shown for the ps-$I_{Ks}$ channel without Mef (black lines) and three for K41C (C, red), W323A (D, blue) after ligand detachment, and Y46C in absence of ligand (E, purple). Dashed lines show average values of RMSF calculated from three simulations. ***p<0.001; **p<0.01 *p<0.05 using an unpaired t-test. GROMACS software was used for RMSF analysis.

The online version of this article includes the following figure supplement(s) for figure 4:

**Figure supplement 1.** Root-mean-square deviations (RMSD) of mefenamic acid from its initial position during MD simulations.

**Figure supplement 2.** Result of loss of mefenamic acid from the binding site.

ps-KCNE1 with significant contributions to DIDS coordination (**Figure 6C**). Although residues identified as critical to mefenamic acid binding and action also generally appear to be important for DIDS, especially W323, K41 was noticeably not as critical to DIDS coordination, since the ΔG values for this residue were significantly smaller compared to those of the Mef (see **Figure 2—figure supplement 2**).

Similar to mefenamic acid, in AMBER force field simulations, DIDS stayed bound to WT ps-$I_{Ks}$, but detached from the W323A mutant during two of three runs of 300 ns duration, and from Y46C

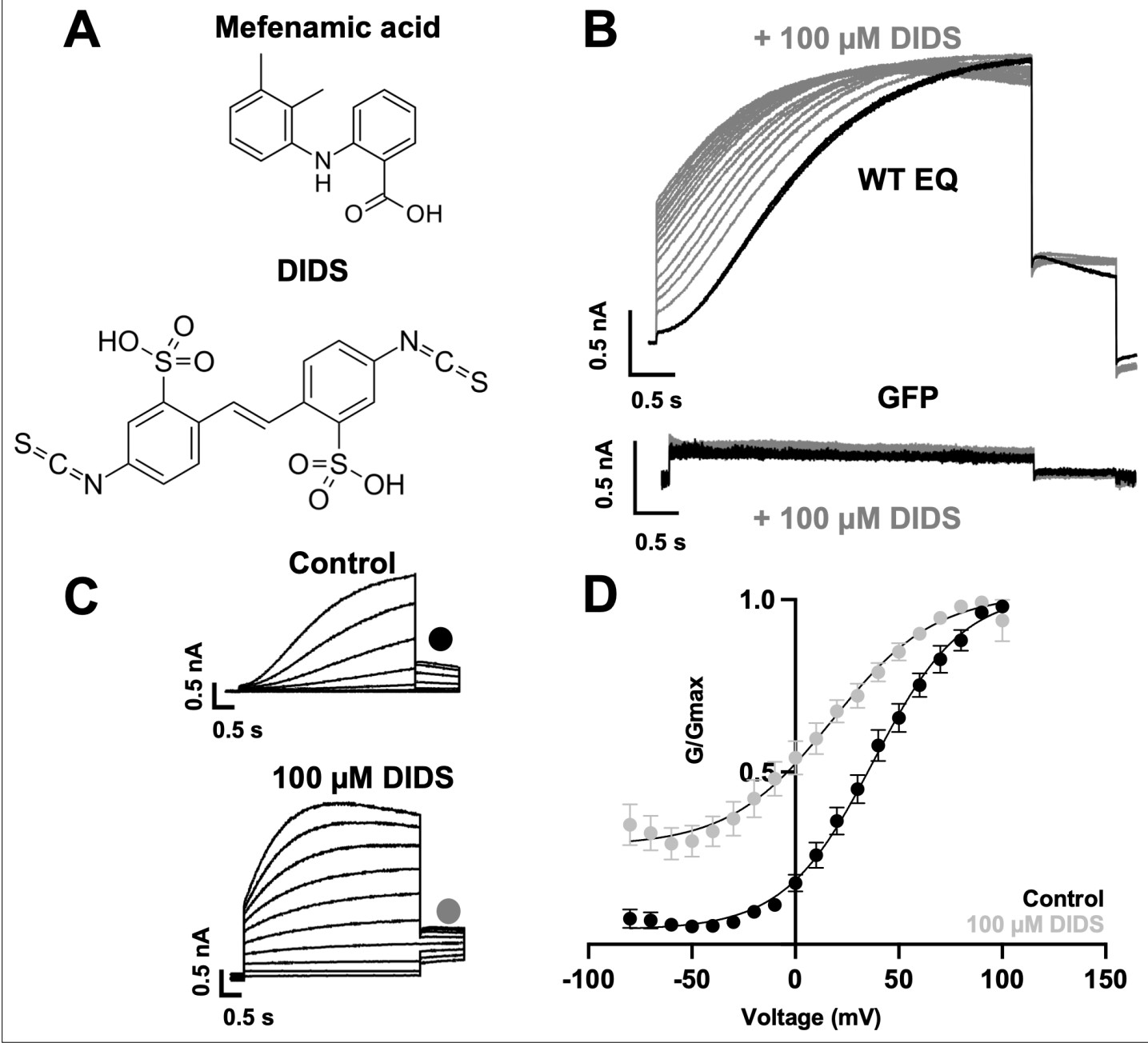

**Figure 5.** Effect of DIDS on $I_{Ks}$. (**A**) Molecular structure of mefenamic acid and DIDS. (**B**) WT EQ current in control (black) and exposed to 100 µM DIDS over time (grey). Pulses were from –80 to +60 mV every 15 s, and current traces are shown superimposed. Lower panel shows no effect on currents from GFP-only transfected cells exposed to 100 µM DIDS over time (grey). (**C**) Current traces from WT EQ in control and presence of 100 µM DIDS as indicated. Pulses were from –80 to +100 mV for 4 s, with a 1 s repolarization to –40 mV. Interpulse interval was 15 s. (**D**) Corresponding G-V plot in control (black) and DIDS (grey) from data as shown in panel C. Boltzmann fits were: WT EQ control (n=8): $V_{1/2}$ = 30.5 mV, $k$=20.3 mV; WT EQ in the presence of DIDS (n=5): $V_{1/2}$ = -16.1 mV, $k$=25.3 mV. Error bars shown are ± SEM.

channels in one of three runs after 240, 239, and 122 ns of simulation, respectively (**Table 6**). Interestingly, DIDS did not unbind from K41C mutant ps-$I_{Ks}$ channels, even after 300 ns of simulation, although the binding pose did change in one run. Calculations of free interaction energy shown in **Table 6** using both MM/PBSA and MM/GBSA methods are of mean values before the detachment of the drug if that occurred. If DIDS detachment did not occur during 300 ns MD runs, the average free energy was calculated over the duration of the entire MD run for that construct. Only DIDS binding to W323A, and to Y46C in one simulation run, showed significantly lower free energy than DIDS binding

**Table 5.** Mean $V_{1/2}$ of activation (mV) and slope values (*k*-factor, mV) in the absence and presence of 100 µM DIDS for fully saturated $I_{Ks}$ channel complexes.
A statistical difference in $V_{1/2}$ compared to control is shown as an p-value determined using an unpaired t-test. NS denotes not significant. Values are shown ± SEM.

| | Control | | | 100 µM DIDS | | | |
|---|---|---|---|---|---|---|---|
| | $V_{1/2}$ | *k-factor* | n | $V_{1/2}$ | *k-factor* | n | p-value |
| WT EQ | 30.5±4.3 | 20.3±0.9 | 8 | -16.1±2.8 | 25.3±1.9 | 5 | <0.001 |
| K41C-EQ | 23.1±3.0 | 20.2±1.8 | 3 | -1.6±3.7 | 24.0±3.6 | 3 | <0.01 |
| L42C-EQ | 55.4±3.3 | 19.9±1.5 | 3 | 21.4±10.7 | 28.5±2.6 | 4 | <0.05 |
| A44C-EQ | 24.1±1.7 | 29.6±3.3 | 4 | 5.6±2.4 | 17.6±1.8 | 3 | <0.05 |
| *Y46A-EQ | 75.2±2.1 | 59.1±5.3 | 5 | 25.8±0.9 | 39.2±1.4 | 5 | <0.0001 |
| EQ-Y148C | 51.8±4.6 | 23.5±1.7 | 4 | 40.0±3.1 | 26.3±1.5 | 5 | NS |
| EQ-W323A | 50.7±3.6 | 23.9±1.6 | 4 | 23.4±4.9 | 24.1±1.3 | 4 | <0.05 |

*An estimation of the activation $V_{1/2}$ was calculated from a right shifted, non-saturating GV curve.

to WT, using both methodologies. The small change in free energy and more stable binding of DIDS to K41C suggest that this residue is not as important as W323 or Y46 for DIDS binding. In longer 500 ns simulations using a CHARMM force field with the channel complexes embedded in lipid, trajectory analysis revealed clusters of DIDS poses in complex with the WT ps-$I_{Ks}$ channel, which are much more closely related than those identified for mefenamic acid (*Figure 6—figure supplement 2*, note the ordinate scale). Still, DIDS remained bound to WT ps-$I_{Ks}$, during all runs throughout 500 ns simulations (*Figure 6—figure supplement 3*, *Video 5*), and to the K41C mutant in two runs (*Video 6*). Both W323A and Y46C mutants unbound during 500 ns simulations (all runs for Y46C, *Figure 6—figure supplement 3*, *Videos 7 and 8*), as was seen with the AMBER force field. These model predictions were tested experimentally using fully saturated mutant $I_{Ks}$ channel complexes, K41C-EQ, L42C-EQ, A44C-EQ, Y46A-EQ, EQ-W323A, EQ-L142C, and EQ-Y148C (*Figure 7A*, *Figure 7—figure supplement 1*). Y46 was identified as an important residue for DIDS binding in MD simulations, but as before, the complexity of the GV curve and low functional expression of the Y46C construct led us to substitute Y46A for this mutation in electrophysiology experiments (*Figure 7* and *Figure 7—figure supplement 1*).

K41C-EQ G-V plots were obtained from peak tail current amplitudes in the absence and presence (*Figure 7B*) of 100 µM DIDS. Unlike with mefenamic acid, but in agreement with the modeling, K41C only partially prevented the action of DIDS. Treatment of K41C-EQ with 100 µM DIDS hyperpolarized the $V_{1/2}$ and changed the shape of the G-V relationship compared to control (*Figure 7B and D*, *Table 5*). The current waveform of K41C-EQ activated more quickly with less sigmoidicity after treatment with DIDS, and tail current decay was slowed as indicated by the normalized response (*Figure 7B and C*). As K41C-EQ remained responsive to DIDS, and L42C responded similarly to K41C-EQ (for K41C-EQ, $\Delta V_{1/2}$ was –24.7 mV, and for L42C-EQ $\Delta V_{1/2}$ was –34 mV), we hypothesized that DIDS binding to $I_{Ks}$ channel complexes may involve KCNE1 residues closer to the transmembrane region. In agreement with this idea, A44C showed little response to 100 µM DIDS both in terms of current waveform and tail current decay. The shift in $V_{1/2}$ was also reduced compared to WT-EQ (A44C-EQ $\Delta V_{1/2}$ = -18.5 mV; WT EQ $\Delta V_{1/2}$ = -46.6mV, *Figure 7C and D*). In Y46A, the effect of DIDS on the current waveform was greatly reduced as was slowing of tail current decay (*Figure 7A and C*). We could only estimate an activation $V_{1/2}$ for Y46A as the G-V curve was right shifted to very positive potentials and non-saturating. Nevertheless, hyperpolarization of the $V_{1/2}$ (*Figure 7D*, *Figure 7—figure supplement 1*) in the presence of DIDS was observed, suggesting that unstable and short-lived binding of the drug to the channel complex was sufficient to interfere with channel gating.

The modeling studies (*Figure 6*) suggested that W323 remained a key residue for $I_{Ks}$ activator sensitivity, and in agreement with this, EQ-W323A was only partially responsive to DIDS. In EQ-W323A, the tail current decay was affected but the slowly-activating current waveform was preserved in the presence of DIDS, supporting the idea that the drug dissociates from the channel complex between

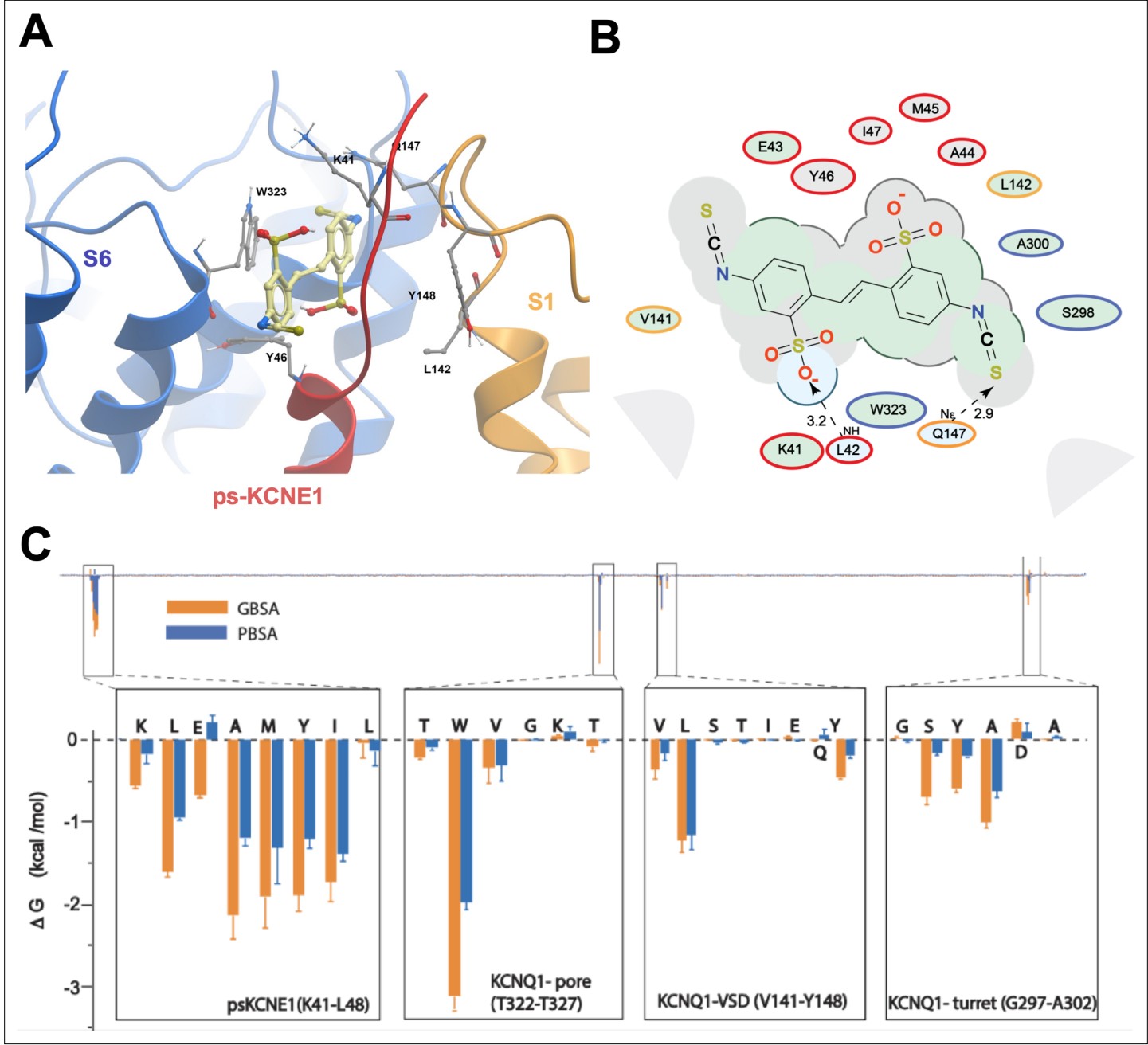

**Figure 6.** Ligand interaction and energy decomposition per amino acid for DIDS binding to ps-$I_{Ks}$. (**A**) Binding pose of DIDS (yellow) in the external region of the ps-$I_{Ks}$ channel complex obtained with molecular docking (side view). The residues of the pore domain are colored in blue, ps-KCNE1 subunit in red, and the voltage-sensor domain of a neighbouring subunit is presented in yellow. (**B**) Ligand interaction map of DIDS with ps-$I_{Ks}$ from molecular docking. Size of residue ellipse is proportional to the strength of the contact. Light grey indicates residues in van der Waals contacts, light green hydrophobic contacts, and light blue are hydrogen bond acceptors. Red borders indicate KCNE1 residues, yellow are KCNQ1 VSD residues, and blue are pore residues. Dashed lines indicate H-bonds. The distance between the residue label and ligand represents proximity. Grey parabolas represent accessible surface for large areas. The 2D diagram was generated by ICM pro software with a cut-off value for hydrophobic contacts 4.5 Å and hydrogen bond strength 0.8. (**C**) Energy decomposition per amino acid for DIDS binding to ps-$I_{Ks}$. Generalized Born Surface Area (MM/GBSA; orange) and Poisson-Boltzmann Surface Area (MM/PBSA; blue) methods were used to estimate the interaction free energy contribution of each residue in the DIDS-bound ps-$I_{Ks}$ complex. The lowest interaction free energy for residues in ps-KCNE1 and selected KCNQ1 domains are shown as enlarged panels (n=3 for each point). Error bars indicate ± SD.

The online version of this article includes the following figure supplement(s) for figure 6:

**Figure supplement 1.** H-bonds formed between DIDS and protein residues in its binding site during MD simulations.

*Figure 6 continued on next page*

*Figure 6 continued*

**Figure supplement 2.** Clustering of MD trajectories based on DIDS conformations.

**Figure supplement 3.** Root-mean-square deviations (RMSD) of DIDS from its initial position during MD simulations.

pulses (*Figure 7A and C*, *Table 6*). However, a significant shift in the $V_{1/2}$ remained ($\Delta V_{1/2}$ = -27.3 mV; *Figure 7D*, *Table 5*). Comparable effects observed in EQ-W323A, A44C-EQ, and Y46A-EQ are consistent with the idea that the DIDS binding pocket involves the N-terminal end of S6 and the deeper extracellular surface of the KCNE1 transmembrane segment. In EQ-Y148C, the $V_{1/2}$ and the shape of the G-V plot were not altered by 100 μM DIDS (*Figure 7D*, *Table 5*), and tail current decay was not markedly slowed (*Figure 7A and C*). This result suggests that unlike mefenamic acid, DIDS binding determinants extend to the S1 region of KCNQ1 and that Y148 is an important contributor.

Notably, as with mefenamic acid, the L142C-EQ G-V plot became voltage-independent after treatment with 100 μM DIDS (data not shown) and so it proved hyper-responsive to both of the $I_{Ks}$ activators, DIDS and mefenamic acid.

## Discussion

Mefenamic acid, a nonsteroidal anti-inflammatory drug (NSAID), and the structurally distinct stilbenes DIDS and SITS, among other compounds, have previously been identified by numerous groups to enhance $I_{Ks}$ currents in various expression systems including canine and guinea-pig ventricular myocytes (*Magyar et al., 2006*; *Toyoda et al., 2006*) as well as heterologous expression systems such as *Xenopus laevis* oocytes, CHO, COS-7, tsA201 and LM cells (*Busch et al., 1994*; *Abitbol et al., 1999*; *Unsöld et al., 2000*; *Toyoda et al., 2006*; *Wang et al., 2020*). The extracellular region of KCNE1 between residues 39 and 44 was found important in mediating the effect of DIDS on $I_{Ks}$ (*Abitbol et al., 1999*) and residue K41, located on the extracellular end of KCNE1, was found to be critical in mediating mefenamic acid's activating effect on the fully saturated $I_{Ks}$ channel complex (*Wang et al., 2020*). Consistent with this idea, when all four WT KCNE1 subunits are replaced with mutant K41C-KCNE1, mefenamic acid up to a concentration of 1 mM is largely ineffective (*Figure 1*). Unlike in the WT complex, there is little change to the current waveform, slope or $V_{1/2}$ of the G-V plot during activator exposure, suggesting that all the drug binding site(s) on the channel complex are impaired, or that the mechanism of action is disabled, or that the mutation causes a combination of these actions. In the present study, we analysed these activator compound actions using molecular modeling approaches, and complementary mutational analysis. Our data describe the formation of a drug binding pocket between the immediate extracellular residues of KCNE1, S1 and pore residues of

**Table 6.** Average free interaction energies of DIDS-bound ps-$I_{Ks}$ complexes calculated according to MM/PBSA and MM/GBSA methods from three independent MD simulation runs of 300 ns duration using the AMBER force field.

Mean values are in kcal/mol, ± SD. For W323A and Y46C mutations, calculations correspond to interval of simulations before the detachment of ligand from the molecular complex. Note that unbinding occurred in W323A in 2 of 3 simulations, and K41C did not unbind but the binding pose shifted.

| Run | Method | ps-$I_{Ks}$ | W323A | Y46C | K41C | A44C | Y148C |
|-----|--------|-------------|-------|------|------|------|-------|
|     | PBSA   | -32.3±6.7   | -25.7±4.9 | -35.5±6.3 | -33.5±4.4 | -29.2±4.2 | -34.1±4.6 |
|     | GBSA   | -35.5±4.6   | -30.2±4.5 | -42.7±5.4 | -41.5±5.8 | -36.8±2.9 | -40.7±4.0 |
| I   |        |             | unbinding after 240 ns |  | changed binding pose after ~140 –150 ns |  |  |
|     | PBSA   | -32.6±4.8   | -25.3±5.5 | -25.7±6.5 | -36.4±4.2 | -29.0±4.0 | -29.3±3.7 |
|     | GBSA   | -36.0±4.8   | -30.0±4.8 | -32.4±6.4 | -31.6±4.0 | -34.0±5.7 | -31.8±4.7 |
| II  |        |             |       | unbinding after ~100 –132 ns |  |  |  |
|     | PBSA   | -29.2±8.1   | -27.1±3.9 | -38.3±4.8 | -35.1±4.3 | -31.2±45.0 | -35.7±5.8 |
|     | GBSA   | -34.0±8.9   | -35.1±4.8 | -42.5±4.7 | -30.8±3.9 | -35.6±5.6 | -40.3±5.9 |
| III |        |             | unbinding after ~240 ns |  |  |  |  |

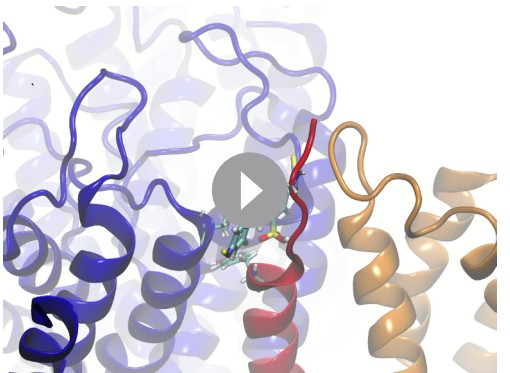

**Video 5.** MD simulations at the molecular level of binding of DIDS to ps-$I_{Ks}$, and K41C-, W323A-, Y46C-ps-$I_{Ks}$ mutants. DIDS binding to ps-$I_{Ks}$. K41, W323, and Y46 side chains are shown.
https://elifesciences.org/articles/87038/figures#video5

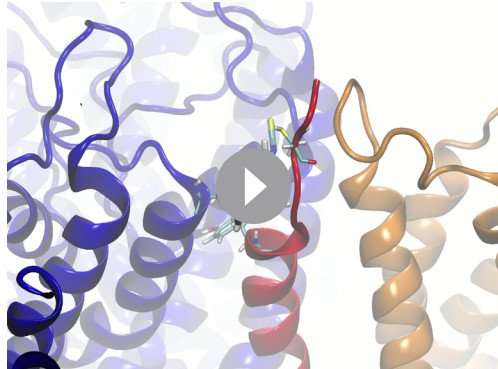

**Video 6.** DIDS binding to K41C ps-$I_{Ks}$. K41C, W323, and Y46 side chains are shown.
https://elifesciences.org/articles/87038/figures#video6

two KCNQ1 subunits, stabilized by the presence of structurally different activator compounds. The effect of mutations that negate the activator compound actions is to destabilize the binding pocket itself, reduce drug binding and limit activator residency time on the channel complex.

## Mefenamic acid binding site in $I_{Ks}$

Using data that suggested an extracellular binding site for mefenamic acid was at the KCNE1-channel interface, MD simulations were used to explore drug-channel interactions further. The drug binding pocket was defined as the extracellular space formed by KCNE1, the domain-swapped S1 helices of one KCNQ1 subunit and the pore/turret region made up of two other KCNQ1 subunits (*Figure 2*; *Video 1*). Docking and molecular simulations identified a binding site made up of several residues across all of these elements (*Figure 2* and *Figure 2—figure supplement 2*, *Table 7*). Mutagenesis revealed that most mutations impacted at least one out of the three mefenamic acid effects: $V_{1/2}$ shift, GV-plot slope change, and current waveform change related to slowed deactivation. However, only three of the mutants impacted all of the mefenamic acid actions (*Figure 3*). These were W323A in KCNQ1, as well as K41C, and A44C (to a lesser extent) in KCNE1, which define key components of the binding pocket for mefenamic acid. That W323 forms an important medial wall of the hydrophobic pocket to which mefenamic acid binds is suggested by data from mutants in which the size of the side chain at this location is systematically made smaller to cysteine and then alanine, which results in diminished effectiveness of the drug, particularly with alanine. This mutation resulted in the detachment of mefenamic acid as well as lower interaction energy of drug-channel complexes during MD simulations (*Figure 4A*; *Video 3*, *Figure 4—figure supplement 1C*) particularly loss of interaction

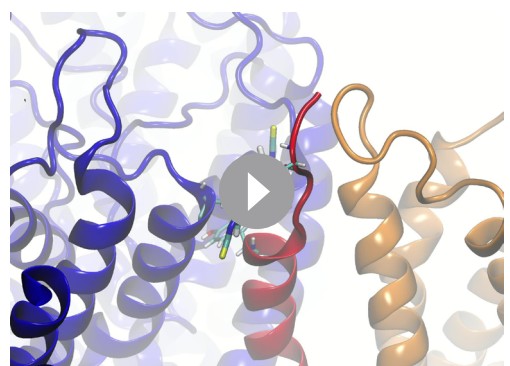

**Video 7.** DIDS binding to W323A ps-$I_{Ks}$. K41, W323A, and Y46 side chains are shown.
https://elifesciences.org/articles/87038/figures#video7

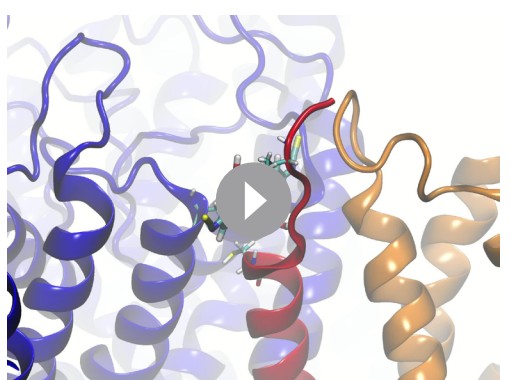

**Video 8.** DIDS binding to Y46C ps-$I_{Ks}$. K41, W323, and Y46C side chains are shown.
https://elifesciences.org/articles/87038/figures#video8

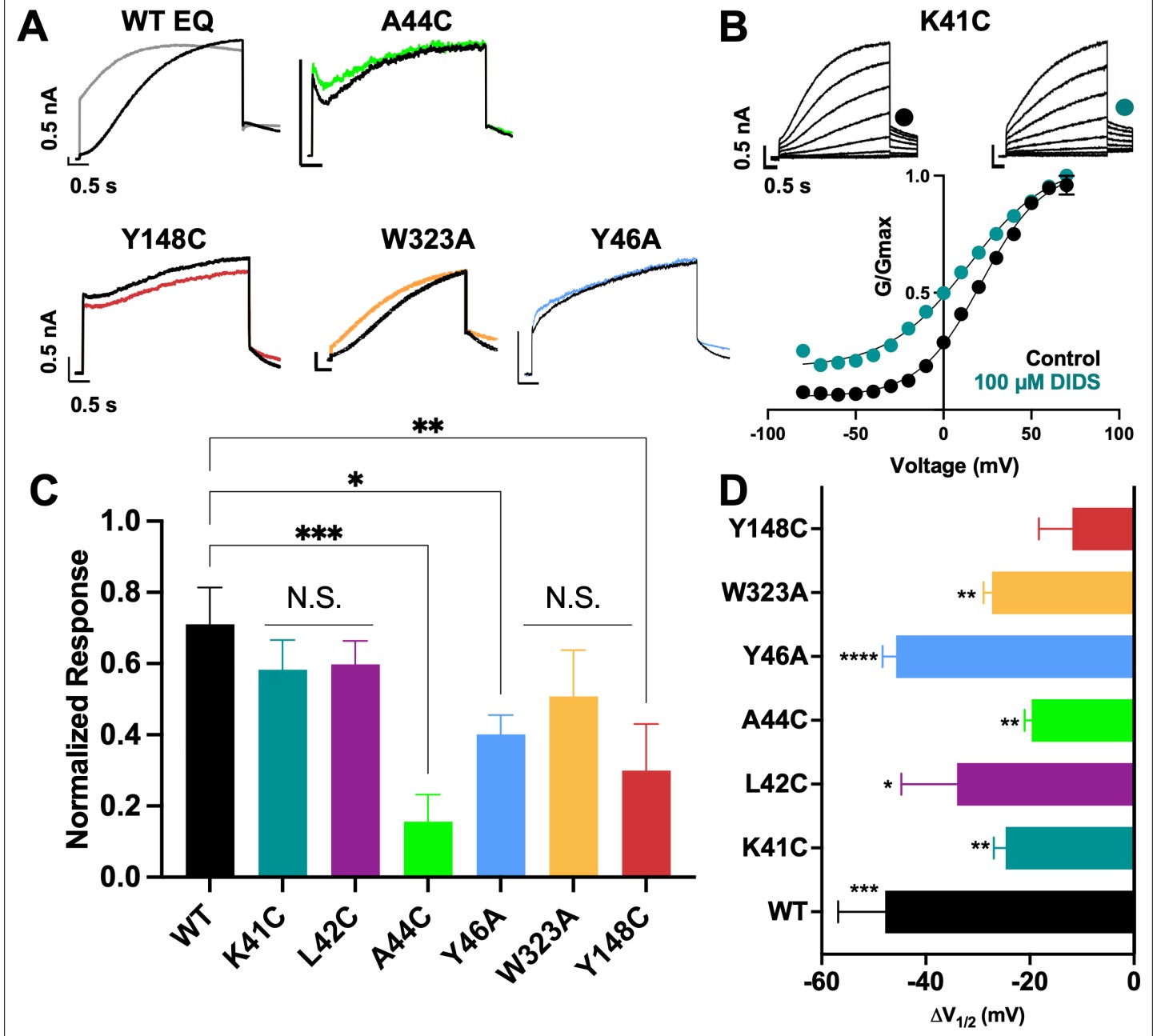

**Figure 7.** Binding site mutants important for the action of DIDS. (**A**) Current traces from WT EQ and key mutants in the absence (control; black) and presence (colors) of 100 µM DIDS. (**B**) Data and G-V plots in control (black) and DIDS (teal). Boltzmann fits were: K41C-EQ control (n=3): $V_{1/2}$ = 23.1 mV, $k$=20.2 mV; K41C-EQ DIDS (n=5): $V_{1/2}$ = -1.6 mV, $k$=24.0 mV. Voltage steps from a holding potential of –80 mV to +70 mV for 4 s, followed by repolarization to –40 mV for 1 s. Interpulse interval was 15 s. Error bars shown are ± SEM. All calibration bars denote 0.5 nA/0.5 s. (**C**) Summary plot of the normalized response to 100 µM DIDS. Data are shown as mean + SEM and *p<0.05, **p<0.01, ***P<0.001 denote significant change in mutant versus WT currents (one-way ANOVA, see Materials and methods). For calculation, see Materials and methods. (**D**) Change in $V_{1/2}$ ($\Delta V_{1/2}$) for WT EQ and each $I_{Ks}$ mutant in control versus DIDS. Data are shown as mean - SEM and unpaired t-test was used, where *p<0.05, **p<0.01, ***p<0.001, ****p<0.0001 indicate a significant change in $V_{1/2}$ comparing control to the presence of drug. n-values for mutants in C and D are stated in *Table 5*.

The online version of this article includes the following figure supplement(s) for figure 7:

**Figure supplement 1.** Current waveform and G-V changes induced by DIDS for all binding site mutants.

**Table 7.** Average free energy of mefenamic acid and DIDS-bound ps-$I_{Ks}$ and mutant complexes.

Values are calculated according to the MM/GBSA method from three independent MD simulation runs using the AMBER force field, and further broken down by residue and channel region. For K41C and W323A, calculations correspond to the interval of simulations before detachment of the ligand from the complex. Values are in kcal/mol. Mutated residues are in bold italics.

| | Residue | ps-$I_{Ks}$ MEF | K41C MEF | Y46C MEF | W323A MEF | ps-$I_{Ks}$ DIDS | A44C DIDS | Y46C DIDS | Y148C DIDS | W323A DIDS |
|---|---|---|---|---|---|---|---|---|---|---|
| Total | | -38.02 | -16.39 | -31.40 | -20.31 | -34.29 | -35.40 | -39.16 | -37.60 | -34.33 |
| Drug | | -19.69 | -8.79 | -15.25 | -11.65 | -17.42 | -17.52 | -21.21 | -18.18 | -19.17 |
| Channel | | -17.91 | -6.52 | -16.61 | -7.55 | -19.21 | -18.40 | -16.94 | -19.21 | -15.10 |
| Ch +Drug | | -37.60 | -15.31 | -31.86 | -19.20 | -36.63 | -35.92 | -38.15 | -37.39 | -34.27 |
| Difference | | -0.42 | -1.08 | 0.46 | -1.11 | 2.34 | 0.52 | -1.01 | -0.21 | -0.06 |
| E1/E3 | K41 | -1.548 | *-0.349* | -1.411 | -0.954 | -0.639 | -0.439 | -1.148 | -0.283 | -0.083 |
| | L42 | -0.970 | -0.895 | -0.943 | -1.193 | -1.061 | -0.798 | -0.165 | -0.320 | -0.587 |
| | E43 | -2.681 | -0.472 | -2.257 | -0.309 | -1.312 | -0.660 | -0.014 | -2.264 | -0.705 |
| | A44 | -1.906 | -0.886 | -2.035 | -1.102 | -1.778 | *-2.062* | -1.027 | -1.061 | -1.780 |
| | M45 | -1.744 | -0.163 | -1.913 | -0.064 | -1.756 | -2.154 | -1.222 | -2.810 | -1.633 |
| | Y46 | -2.391 | -0.121 | *-2.702* | -0.083 | -1.985 | -1.797 | *-1.245* | -2.377 | -1.603 |
| | I47 | -0.771 | -0.136 | -0.412 | -0.203 | -2.036 | -1.993 | -1.004 | -0.621 | -1.528 |
| | L48 | -0.010 | -0.003 | -0.025 | 0.005 | -0.045 | -0.100 | -0.233 | -0.081 | -0.035 |
| Sum | | -12.02 | -3.03 | -11.70 | -3.90 | -10.61 | -10.00 | -6.06 | -9.82 | -7.79 |
| Pore | T322 | -0.125 | -0.047 | -0.091 | -0.968 | -0.200 | -0.213 | -0.127 | -0.152 | -0.410 |
| | W323 | -2.159 | -0.736 | -2.226 | *-0.932* | -3.312 | -3.395 | -2.279 | -3.566 | *-1.590* |
| | V324 | -0.644 | -0.037 | -0.123 | -0.047 | -0.172 | -0.210 | -0.111 | -2.016 | -0.063 |
| | G325 | -0.038 | -0.006 | -0.006 | 0.017 | -0.004 | 0.000 | 0.003 | -0.048 | 0.032 |
| | K326 | -0.185 | -0.015 | -0.024 | 0.008 | 0.038 | 0.058 | 0.091 | 0.000 | 0.118 |
| | T327 | -0.130 | -0.015 | 0.020 | 0.027 | -0.040 | -0.022 | -0.018 | -1.298 | -0.014 |
| Sum | | -3.28 | -0.86 | -2.49 | -1.89 | -3.69 | -3.78 | -2.44 | -7.08 | -1.93 |
| S1 | V141 | -0.061 | -0.014 | -0.247 | -0.006 | -0.569 | -0.411 | -0.660 | -0.076 | -0.504 |
| | L142 | -0.861 | -0.434 | -1.052 | -0.267 | -1.525 | -0.822 | -1.285 | -0.460 | -1.036 |
| | S143 | -0.028 | -0.004 | -0.014 | -0.063 | -0.008 | -0.006 | -0.104 | -0.026 | -0.017 |
| | T144 | -0.091 | -0.006 | -0.242 | -0.526 | -0.015 | -0.122 | -0.982 | -0.002 | -0.001 |
| | I145 | 0.004 | 0.005 | 0.002 | 0.009 | 0.003 | -0.329 | -0.067 | -0.009 | -0.001 |
| | E146 | 0.000 | 0.048 | 0.056 | 0.114 | 0.028 | -0.014 | -1.601 | 0.113 | 0.013 |
| | Q147 | -0.230 | -0.365 | 0.006 | -0.244 | 0.012 | -0.143 | -1.061 | -0.974 | -0.054 |
| | Y148 | -0.565 | -1.556 | 0.006 | -0.128 | -0.428 | -0.336 | 0.002 | *-0.022* | -0.223 |
| Sum | | -1.83 | -2.32 | -1.49 | -1.11 | -2.50 | -2.18 | -5.76 | -1.40 | -1.82 |
| Turret | G297 | -0.024 | -0.001 | -0.010 | -0.027 | -0.016 | -0.027 | -0.089 | -0.001 | -0.076 |
| | S298 | -0.320 | -0.163 | -0.419 | -0.427 | -0.791 | -0.848 | -1.107 | -0.216 | -1.026 |
| | Y299 | -0.103 | -0.046 | -0.104 | -0.052 | 0.718 | -0.598 | -0.563 | -0.167 | -0.673 |
| | A300 | -0.618 | -0.183 | -0.636 | -0.431 | -1.130 | -1.112 | -0.751 | -0.652 | -1.093 |
| | D301 | 0.311 | 0.075 | 0.248 | 0.299 | 0.231 | 0.172 | -0.181 | -0.136 | -0.683 |
| | A302 | -0.024 | -0.003 | -0.011 | -0.007 | -0.011 | -0.020 | 0.009 | -0.008 | -0.015 |
| Sum | | -0.78 | -0.32 | -0.93 | -0.65 | -2.40 | -2.43 | -2.68 | -0.91 | -3.57 |

strength with KCNE1 (*Table 7*). Likewise, MD simulations revealed detachment of mefenamic acid and reduced interaction energy values for K41C-Mef-$I_{Ks}$ complexes (*Video 2*, *Figure 4*) again accompanied by loss of interaction strength with KCNE1 (*Table 7*). The drug also detached from the A44C complex, another mutant with a reduced response in terms of changes in waveform, $V_{1/2}$ and slope. In contrast, while the free interaction energy of the mutant Y46C channel was significantly decreased in comparison to the WT channel complex (*Figure 4A*), detachment of mefenamic acid from the mutant channel during 300 ns MD simulations was not observed (*Table 4*). However, when placed in a lipid environment mefenamic only remained bound in three out of five 500 ns simulations with Y46C, suggesting a more dynamic interaction in this model (*Figure 4—figure supplement 1D*, *Video 4*). Comparison of energies at specific residues shows a loss of interaction with Q147 and Y148 in S1 with this mutant and a small shift towards the turret at S298 and A300 (*Figure 8A*; *Table 7*, *Figure 8—source data 1*). While we were not able to obtain good electrophysiological data from Y46C, data from Y46A suggested that mefenamic acid still had potent actions on this mutant. Overall, the combination of binding studies and electrophysiological data indicate that Y46 was not as important for mefenamic acid binding and action as other KCNE1 N-terminal residues.

## Common binding site for $I_{Ks}$ activators, mefenamic acid and DIDS

Although it was suggested that DIDS and mefenamic acid have the same binding site (*Abitbol et al., 1999*), the extent of overlap was unknown. Initially, we examined the effect of DIDS on WT EQ. Consistent with previous studies (*Abitbol et al., 1999*; *Bollmann et al., 2020*) we confirmed that 100 µM DIDS enhanced WT EQ activity (*Figure 5B*) with a $V_{1/2}$ shift of –46.6 mV and a decrease in the slope of the G-V curve (*Table 5*). As most $I_{Ks}$ activators are dependent on the KCNQ1-KCNE1 stoichiometry, this more potent effect of DIDS seen in our study may be explained by the higher dose we used and our KCNE1-KCNQ1 linked channel constructs, which ensured fully KCNE1-saturated complexes. Furthermore, due to the large size and complex folds found on the surface of oocytes, higher drug concentrations than those used for cultured cells are often required in order to facilitate a similar effect in both expression systems (*Kvist et al., 2011*). All previous studies utilized *Xenopus laevis* oocytes, whereas in this study transiently transfected tsA201 cells were used.

The results of in-silico experiments, including some binding properties and stability of some mutations, were only partially validated by electrophysiology data, which might be explained by the limitations of the applied methods. Nevertheless, docking analyses revealed that both mefenamic acid and DIDS bind to the same general extracellular inter-subunit interface with some differences in key residues revealed by the electrophysiology data. Mutation of Y148 in KCNQ1 to a cysteine (Y148C) or Y46 in KCNE1 to an alanine (Y46A) was found to reduce the effects of 100 µM DIDS, particularly in the case of Y148C (*Figure 7*). These data suggest that DIDS resides deeper in the binding pocket formed by KCNE1 residues and the KCNQ1 pore/S6, and so was less dependent on the side chain of K41 to retain the activator on the channel complex (compare *Video 1* and *Video 5*). DIDS associated more strongly with the pore of the Y148C mutant, particularly with V324 and T327, and less across the ps-KCNE1, S1 and turret regions (*Table 7*). The Y46C mutation resulted in shifts away from ps-KCNE1 and towards the S1 domain (T144, E146 and Q147). In the case of A44C and DIDS, the changes were more subtle, with stronger interactions moving from central residues in KCNE1 (K41, L42, and E43) and S1 (V141 and L142; colored grey in *Figure 8B*, cf. *Figure 8—source data 2*) to more peripheral residues (I145, E146, and Q147) and lower in the KCNE TMD (A44 and M45; colored magenta in *Figure 8B*).

## Proposed mechanism of action for mefenamic acid and DIDS

Molecular modeling and docking revealed that mefenamic acid and DIDS induce conformational changes in the channel upon binding, to shape a binding pocket formed by residues from the external S1 domain, KCNE1 and the pore domain of $I_{Ks}$ (*Figure 2*, *Videos 1 and 5*). This cryptic binding pocket is not detectable in the absence of the drug (*Figure 4B*), which suggests that it has been induced in a similar manner to previous reports of toxin interactions with KcsA-Kv1.3 that induce conformational changes in both the toxin and the channel structure to generate a high-affinity binding site (*Lange et al., 2006*; *Zachariae et al., 2008*). Indeed, analysis of the binding site before and after mefenamic acid unbinds shows the involved residues in the channel complex moving into the space vacated by

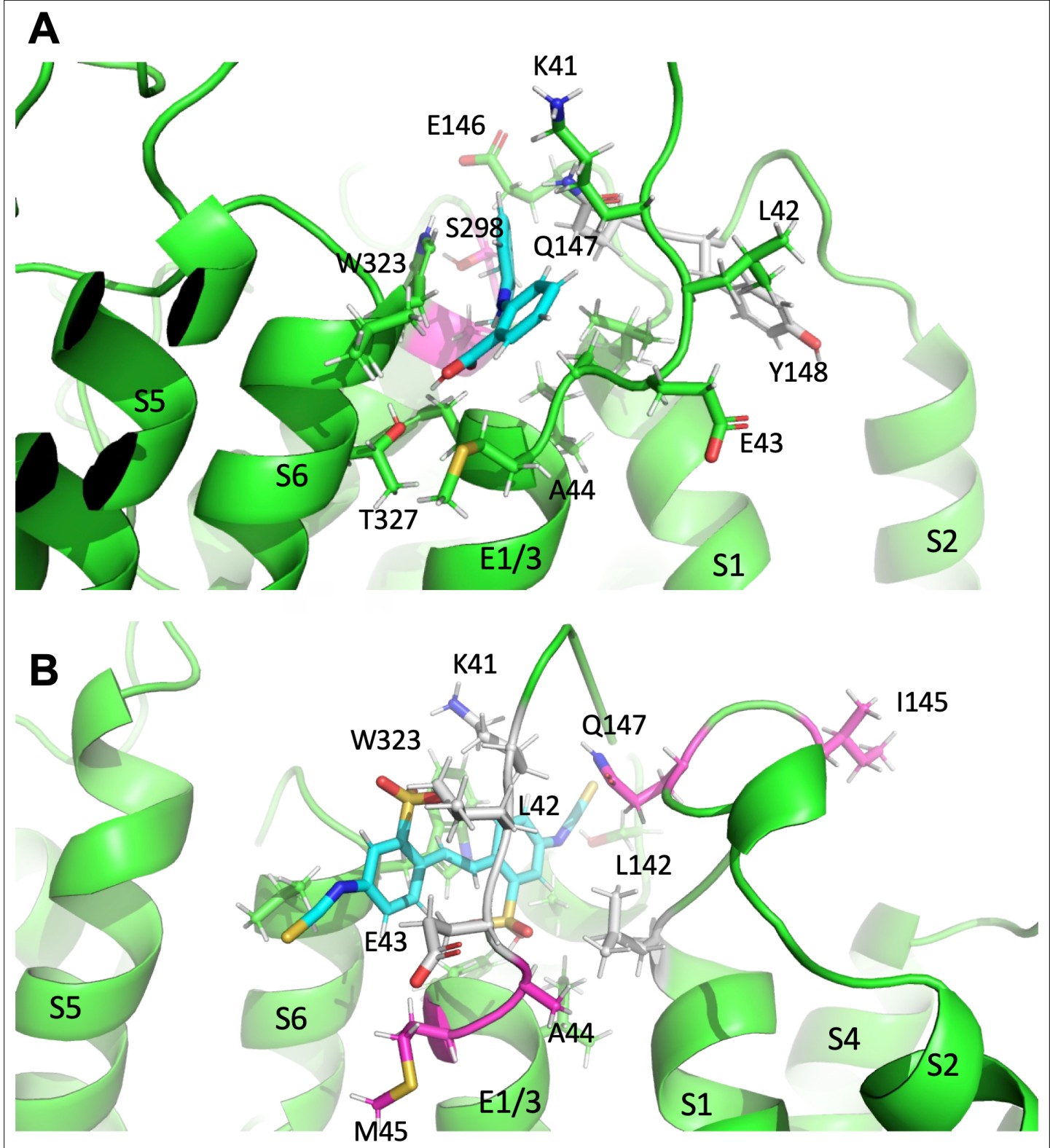

**Figure 8.** Subtle differences in activator binding to Y46C- and A44C-ps-$I_{Ks}$ mutant channels. (**A**) Mefenamic acid bound to the ps-$I_{Ks}$ channel. Residues that are part of the binding site are shown in stick format colored green, except those residues that were important in the WT channel that had reduced ΔG in Y46C (in grey). Residues that had slight increases in ΔG values in Y46C are shown in magenta (S298, A300). Mefenamic acid is in cyan. See *Figure 8—source data 1*. (**B**) DIDS bound to the ps-$I_{Ks}$ channel. Residues that are part of the binding site are shown in stick format colored green except those residues that were important in the WT channel that had reduced ΔG in the A44C mutant (in grey). Residues that had slight increases in ΔG values in

*Figure 8 continued on next page*

*Figure 8 continued*

A44C are shown in magenta. DIDS is in cyan. W323 may be seen behind DIDS. See *Figure 8—source data 2*. Images were made with the PyMOL Molecular Graphics System, Version 2.0 Schrödinger, LLC.

The online version of this article includes the following source data for figure 8:

**Source data 1.** Source data for *Figure 8A*.

**Source data 2.** Source data for *Figure 8B*.

mefenamic acid (*Figure 4—figure supplement 2*, *Video 9*), a collapse of the pocket created by the drug-channel interactions.

Given the high modeled binding energy of the channel-drug activated state complexes, ~−39 kcal/mol for mefenamic acid and ~−35 kcal/mol for DIDS, but the relatively low affinities, given the micromolar concentrations required experimentally, we can imagine a dynamic complex where mefenamic acid and DIDS bind/unbind from $I_{Ks}$ at high frequency. Slowed deactivation may therefore be the result of these rapid binding/unbinding interactions slowing the dissociation of the S1/KCNE1/pore domain/drug complex by either providing steric hindrance to dissociation or by stabilizing the activated complex. MD simulations suggest the latter is most likely the case. Cross-linking studies have previously shown that placing cysteines at key locations in the KCNE1 N-terminus, the top of S1 and in S6 can lead to disulfide bond formation and slowing or elimination of deactivation (*Chung et al., 2009*) similar to what we observe when $I_{Ks}$ is exposed to mefenamic acid and DIDS. Our data indicate that mutation of residue W323 to an alanine would not only destabilize the external S1/KCNE1/pore domain interface (*Figure 4D*, *Table 1*) but also eliminate direct hydrophobic contacts which normally occur between the W323 side chain and mefenamic acid, thus facilitating drug dissociation (*Figure 2C*, *Video 3*).

A reduced interaction with S1 could also conceivably curtail the ability of $I_{Ks}$ activators to slow dissociation of the activated complex and restore faster deactivation rates such that there is no longer an enhanced step current in the presence of the drug, as seen in mutants K41C and A44C for mefenamic acid (*Figures 1A and 3A*) and A44C, Y46A, W323A, and Y148C for DIDS (*Figure 7A*). Similarly, electrostatic interactions of residue K41, which acts as a lid for the binding pocket for mefenamic acid (*Video 2*), could help stabilize the S1/KCNE1/pore complex by its contacts with drug, which in its turn links the static pore domain to the dynamic voltage-sensor. Mutation of residue K41 to a cysteine may prevent stable $I_{Ks}$ activator binding to the channel by increasing the fluctuations of the external KCNE1 region (*Figure 4C*) and by reducing the contacts with the drug (*Figure 2C*). In simulations, mefenamic acid was seen dissociating from mutant K41C, A44C, and W323A channel complexes much more often compared to wild type $I_{Ks}$, and this explains why the electrophysiological effects of $I_{Ks}$ activators were prevented by amino acid substitutions at these locations (*Videos 2 and 3*). Shifts in, and particularly reductions in ps-KCNE1 association for both drugs (*Table 7*) that lead to loss of efficacy suggest that it is the interactions with KCNE1 and S1 that are key to maintaining the activated state. This is similar to the proposal for tight interactions between S1 and KCNE3 in maintaining constitutive activity in KCNE3-associated channels (*Kasuya and Nakajo, 2022*).

In close proximity to this drug-binding pocket are known gain-of function mutations in S1, S140G and V141M, which have previously been reported to slow deactivation in the presence of KCNE1 (*El Harchi et al., 2010*; *Peng et al., 2017*). As their current waveforms are qualitatively similar to those seen after exposure of WT EQ to $I_{Ks}$ activators, we propose that the same mechanism may underlie the effects of mefenamic acid, DIDS and these $I_{Ks}$ S1 mutations. In addition, we find that mutation of the neighboring residue, L142,

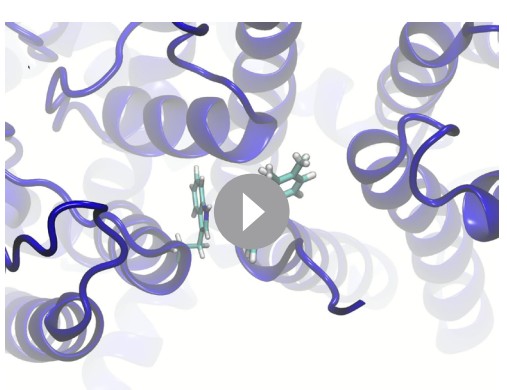

**Video 9.** Binding pocket fluctuations before and after exit of mefenamic acid. W323A and the backbone of ps-KCNE1 (residues 41–44) gradually appear ~100 ns. Frames before mefenamic acid detachment are white, and after detachment red.

https://elifesciences.org/articles/87038/figures#video9

also produces current waveforms and G-V plots which mirror those seen when WT EQ is treated with 100 μM mefenamic acid (*Figure 3—figure supplement 2*) or DIDS. In the cryo-EM structure of KCNQ1-KCNE3 (*Sun and MacKinnon, 2020*), residue L142 interacts with KCNE3, whereas V141 interacts with both KCNE3 and the pore domain. This suggests that both the V141M and L142C mutations could directly alter the interaction of the S1 domain with the pore and/or the position or movement of KCNE1. The co-evolved interface between the extracellular end of S1 and the pore domain is thought to be important for bracing the VSD, to allow efficient force transmission to the pore (*Lee et al., 2009*), but can also impact permeation, as the S140G and V141M mutations also enhance rubidium permeation through $I_{Ks}$ complexes (*Peng et al., 2017*). The importance of this S1-pore coupling to channel function is supported by mutational analyses of S1 residues (*Chen et al., 2003*; *Hong et al., 2005*; *Wang et al., 2011*; *Campbell et al., 2013*) as well as the L142C mutation examined in this study (*Figure 3—figure supplement 2*), which all display current waveforms with instantaneous onset. Incidentally, A300, a residue in the pore region of KCNQ1, which also interacts with both mefenamic acid and DIDS in MD simulations (*Figure 2—figure supplement 2*, *Figure 6*), has also been implicated in the same gain-of-function cleft as S140G and V141M (*Smith et al., 2007*). The A300T mutant has a ~–20 mV shift in $V_{1/2}$ of activation and faster rates of activation (*Bianchi et al., 2000*), once again showing how a mutation can mirror the effects of the two activators studied here.

## Conclusion

Binding of mefenamic acid and DIDS to the extracellular end of KCNE1 and the KCNQ1 S6 and S1 helices is facilitated by a number of key residues. Residue K41 acts as a 'lid' holding mefenamic acid in place, while residue W323 impacts the size of the binding pocket. Size reduction mutations of either residue destabilize mefenamic acid binding and ultimately lead to drug detachment from the channel complex. This explains why, when all four $I_{Ks}$ subunits are mutated, as in the case of K41C-EQ, EQ-W323A, and A44C-EQ, little to no effect of the drug is seen. The larger drug, DIDS, interacts with many of the same residues but those deeper in the pocket appear more important than for mefenamic acid. Furthermore, the qualitative similarities between the S1 mutant channel, EQ-L142C and WT EQ in the presence of 100 μM mefenamic acid suggest that $I_{Ks}$ activators most likely cause their effects by modulating interactions between the S1 helix, pore turret, KCNE1 and the S6 helix. Upon binding, both DIDS and mefenamic acid induce conformational changes in an occult binding pocket and stabilize the S1/KCNE1/pore complex, which ultimately slows deactivation. The results indicate that this extracellular inter-subunit interface forms a generalized binding site which different drugs can access and through which induce common effects on channel activation and deactivation. The presence of such a binding site and the variable nature of $I_{Ks}$ complex composition may serve as starting points for future drug development projects targeted at discovering therapeutically-useful $I_{Ks}$ agonists.

## Materials and methods
### Molecular docking and molecular dynamic simulations

A model of the $I_{Ks}$ channel complex – termed pseudo-KCNE1-KCNQ1 (ps-$I_{Ks}$) – was created based on the cryogenic electron microscopy (cryo-EM) structure of KCNQ1-KCNE3 (PDB ID: 6v01; *Sun and MacKinnon, 2020*). In this structure, the extracellular residues of KCNE3, R53-Y58, were substituted with homologous KCNE1 residues D39-A44. Conformational sampling was then performed for substituted residues and the lowest free energy conformations were selected for subsequent docking experiments applying a four-dimensional (4D) docking approach which accounts for the flexibility of the receptor site configuration (*Bottegoni et al., 2009*). After docking, a receptor ensemble with multiple conformations of the putative binding site region was created via another round of conformational sampling for the external part of ps-KCNE1 and its KCNQ1 neighborhood (8 Å cut-off distance) and generated conformations were used for a new docking iteration. A ligand-channel conformation with the lowest free energy was chosen from the final docking iterations. The docking and conformational sampling as well as substitution of amino acids were performed with ICM-pro 3.8 software (*Neves et al., 2012*). A schematic representation of the general workflow for ps-$I_{Ks}$ model construction and drug docking can be found in *Figure 2—figure supplement 1*.

The coordinates of mefenamic acid-bound ps-$I_{Ks}$ channel complexes with the lowest free energy were then used for two sets of MD simulations with AMBER and CHARMM force fields (see below).

Considering the complexity of the binding site located on the periphery of the pore, ps-KCNE1 and VSD, docking and subsequent MD simulations were performed for only one binding site (*Figure 2D*). To keep the voltage-sensing domain (VSD) in its activated state conformation, we restrained the PIP2 molecules in their cryo-EM positions with a force constant of 1000 kJ/mol/nm$^2$ during MD simulations.

Three 300 ns duration simulations were performed in a water environment with AMBER20 using a ff14SB force field for protein and GAFF/AM1-BCC scheme for the ligand parameterization and calculation of the atomic point charges (*Jakalian et al., 2002*; *Case et al., 2005*; *Maier et al., 2015*). The complex was solvated with TIP3P water models and Na$^+$/Cl$^-$ at 100 mM concentration. The system was minimized and equilibrated in the NVT and NPT ensembles for 10 ns, gradually releasing spatial restraints from the backbone and sidechains. During the last 5 ns only backbone atoms were restrained with a force constant of 50 kJ/mol/nm$^2$. The same procedure was used for equilibration of mutant complexes after introduction of mutations using ICM-pro 3.8 software. We used the Langevin thermostat with a collision frequency of 2 ps$^{-1}$, a reference temperature of 310 K, and Monte Carlo barostat with reference pressure at 1 bar (*Oliver et al., 1997*; *Wu et al., 2016*). The long-range electrostatic interactions with a cut-off at 10 Å were treated with the Particle Mesh Ewald (PME) algorithm. Bonds involving only hydrogens were constrained with the SHAKE algorithm and a 2 fs integration time step was used.

Five 500 ns duration simulations were performed using a CHARMM36m force field and ps-$I_{Ks}$ complex inserted into a POPC membrane. The same MD parameters described in our previous study were used for this set of simulations (*Willegems et al., 2022*). Trajectories from MD simulations were clustered with TTclust based on ligand and its binding site RMSD. The elbow method was used to find the optimum number of clusters. From each cluster, a centroid with the lowest RMSD to all other ligand conformations in the cluster was selected as a representative structure. GROMACS-2021.4 was used for RMSD, RMSF, and H-bond calculation. RMSD was calculated from the initial position of the ligand after the least squares fit alignment of the protein backbone. For RMSF calculations only backbone and C-beta atoms were used.

Trajectories from MD simulations with a CHARMM36m force field are available at https://doi.org/10.5281/zenodo.8226585. The 2D diagrams and other molecular visualizations were generated by ICM-pro and VMD software. The MM/PBSA and MM/GBSA methods were used to calculate the free energy of binding (*Miller et al., 2012*). We collected 1000 snapshots with equal intervals (every 0.3 ns) if ligands did not leave the binding site during 300 ns.

## Reagents and solutions

DIDS (Sigma-Aldrich, Mississauga, ON, Canada) was prepared as a 50 mM stock solution dissolved in 100% dimethyl sulfoxide. Stock DIDS solution was diluted in control whole-cell bath solution to obtain a final DIDS concentration of 100 μM which was perfused onto mammalian cells for whole-cell experiments. All other reagents and solutions were prepared as described (*Wang et al., 2020*). Mefenamic acid (Tocris Bioscience, Oakville, ON, Canada) was used to activate $I_{Ks}$ at concentrations of 100 μM and 1 mM.

## Molecular biology, cell culture and whole cell patch clamp

tsA201 transformed human embryonic kidney 293 cells were purchased directly from Sigma-Aldrich. After culturing, they were plated for whole-cell experiments and subsequently transfected with Lipofectamine2000 (*Murray et al., 2016*; *Westhoff et al., 2019*; *Wang et al., 2020*). All mutations were first generated using site-directed mutagenesis and Pfu turbo, then sequence confirmed. Whole-cell experiments were conducted 24–48 hr post transfection. For wild type (WT) EQ and mutant x-EQ-Y (where 'x' denotes a KCNE1 mutation and 'y' denotes a KCNQ1 mutation; for example, K41C-EQ, EQ-W323A), cells were transfected with a linked KCNE1 and KCNQ1 cDNA (2–3 μg was used) which assembles as a fully saturated 4:4 ratio of KCNE1 to KCNQ1. All constructs were also co-transfected with 0.8 μg of GFP to allow transfected cells to be identified. Data were obtained using an Axopatch 200B amplifier, Digidata 1440 A digitizer and pCLAMP 11 software (Molecular Devices, LLC, San Jose, CA).

## Data analysis

Conductance-voltage (G-V) relationships were obtained from the normalized peak of the initial tail current (G/Gmax) and plotted against the corresponding voltage. G-V plots were fitted with a Boltzmann sigmoid equation to obtain the voltage at half-maximal activation ($V_{1/2}$) and slope ($k$) values (*Tables 1, 3 and 5*). For each EQ mutant, the change in activation $V_{1/2}$ ($\Delta V_{1/2} = V_{1/2}$ in the presence of drug-$V_{1/2}$ control) was also determined (*Figures 3E and 7D*). In some cases, the foot of the G-V curve did not reach 0 (essentially in the presence of drug) and as a result, the $V_{1/2}$ was read directly from the normalized plots at the voltage point where the curve crossed the 0.5 value on the y coordinate. Slowing of tail current decay was used as a measure of mefenamic acid and DIDS impact on WT and mutated EQ channel complexes. Specifically, the peak to end difference currents were calculated by subtracting the minimum amplitude of the deactivating current from the peak amplitude of the deactivating current. The difference current in mefenamic acid or DIDS was normalized to the maximum control (in the absence of drug) difference current and subtracted from 1.0 to obtain the normalized response (*Figures 3D and 7C*). Data files collected during this study and used in the preparation of results and figures are available at https://doi.org/10.5281/zenodo.8226585.

GraphPad Prism 9 (GraphPad Software, San Diego, CA) was used to analyze all the data. Where applicable, unpaired t-test or one-way ANOVA followed by the Fisher's least significant difference (LSD) test was used to determine statistical significance. In all figures ****, ***, **, * denotes a significance where $p < 0.0001$, $p < 0.001$, $p < 0.01$ and $p < 0.05$, respectively. All data in the figures and tables are shown as mean ± SD or SEM. Bar graphs showing mean $\Delta V_{1/2}$ (*Figures 3E and 7D*) were generated by calculating changes in $V_{1/2}$ induced by drug treatment vs. control in separate cells. $\Delta V_{1/2}$ values reported in the Results were calculated from the mean $V_{1/2}$ values shown in *Tables 1 and 3*.

## Acknowledgements

We thank Fariba Ataei for her assistance in cell culture and for making mutants. This research was funded by Natural Sciences and Engineering Research Council of Canada (grant #RGPIN-2022–03021), Canadian Institutes of Health Research (#PJT-175024) and Heart and Stroke Foundation of Canada (#G-21–0031566) grants to DF, and grants from the Volkswagen Foundation (#AZ86659 and AZ 92111) to VV. MC holds an NSERC CGS-M scholarship. YW holds a CIHR– Vanier CGS scholarship.

## Additional information

### Funding

| Funder | Grant reference number | Author |
| --- | --- | --- |
| Natural Sciences and Engineering Research Council of Canada | RGPIN-2022-03021 | David Fedida |
| Canadian Institutes of Health Research | PJT-175024 | David Fedida |
| Heart and Stroke Foundation of Canada | G-21-0031566 | David Fedida |
| Volkswagen Foundation | AZ86659 | Vitya Vardanyan |
| Volkswagen Foundation | AZ92111 | Vitya Vardanyan |

The funders had no role in study design, data collection and interpretation, or the decision to submit the work for publication.

### Author contributions

Magnus Chan, Conceptualization, Data curation, Formal analysis, Writing – review and editing; Harutyun Sahakyan, Conceptualization, Data curation, Software, Formal analysis, Investigation, Methodology, Writing – original draft, Writing – review and editing; Jodene Eldstrom, Conceptualization, Data curation, Formal analysis, Supervision, Funding acquisition, Investigation, Methodology, Writing – original draft, Project administration, Writing – review and editing; Daniel Sastre, Data curation,

Formal analysis; Yundi Wang, Conceptualization, Data curation; Ying Dou, Data curation; Marc Pourrier, Formal analysis, Supervision, Investigation, Methodology; Vitya Vardanyan, Conceptualization, Data curation, Supervision, Writing – review and editing; David Fedida, Conceptualization, Data curation, Formal analysis, Supervision, Funding acquisition, Validation, Investigation, Methodology, Writing – original draft, Project administration, Writing – review and editing

## Author ORCIDs

Harutyun Sahakyan ⓘ https://orcid.org/0000-0003-3750-8118
Vitya Vardanyan ⓘ http://orcid.org/0000-0001-6731-036X
David Fedida ⓘ http://orcid.org/0000-0001-6797-5185

Reviewer #1 (Public Review): https://doi.org/10.7554/eLife.87038.3.sa1
Reviewer #2 (Public Review): https://doi.org/10.7554/eLife.87038.3.sa2
Reviewer #3 (Public Review): https://doi.org/10.7554/eLife.87038.3.sa3
Author Response: https://doi.org/10.7554/eLife.87038.3.sa4

---

## Additional files

### Supplementary files

• MDAR checklist

### Data availability

All original electrophysiological data files summarized in figures and figure supplements and CHARMM trajectory data are available at https://doi.org/10.5281/zenodo.8226585.

The following dataset was generated:

| Author(s) | Year | Dataset title | Dataset URL | Database and Identifier |
|---|---|---|---|---|
| Chan M, Sahakyan H, Eldstrom J, Sastre D, Wang Y, Dou Y, Pourrier M, Vardanyan V, Fedida D | 2023 | A generic binding pocket for small molecule IKs activators at the extracellular inter-subunit interface of KCNQ1 and KCNE1 channel complexes | https://doi.org/10.5281/zenodo.8226585 | Zenodo, 10.5281/zenodo.8226585 |

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

# Appendix 1

**Appendix 1—key resources table**

| Reagent type (species) or resource | Designation | Source or reference | Identifiers | Additional information |
|---|---|---|---|---|
| Gene (*Homo sapiens*) | KCNQ1 | GenBank | HGNC:HGNC:6294 | Gene ID: 3784 |
| Gene (*Homo sapiens*) | KCNE1 | GenBank | HGNC:HGNC:6240 | Gene ID: 3753 |
| Strain, strain background (include species and sex here) | n/a | n/a | n/a | n/a |
| Genetic reagent (include species here) | n/a | n/a | n/a | n/a |
| Cell line (*Homo-sapiens*) | tsa201 | Sigma-Aldrich | Cat # CB_96121229 | Transformed human embryonic kidney 293 cells. The cells have been eradicated from mycoplasma at ECACC. The identity of tsA201 and 293 has been confirmed by STR profiling. |
| Transfected construct (synthetic) | KCNQ1 in pcDNA3 | This paper | | KCNQ1 DNA in pcDNA3 vector. |
| Transfected construct (synthetic) | KCNE1 in pcDNA3 | This paper | | KCNE1 DNA in pcDNA3 vector. |
| Biological sample (include species here) | n/a | n/a | n/a | n/a |
| Antibody | n/a | n/a | n/a | n/a |
| Recombinant DNA reagent | pcDNA3 | Invitrogen | Cat # V79020 | pcDNA3.1 $^{(+)}$ Mammalian Expression Vector |
| Sequence-based reagent | K41C_F | This paper | PCR primers | CCGCAGCGGTGACGGCTGCCTGGAGGC |
| Sequence-based reagent | K41C_R | This paper | PCR primers | GTAGAGGGCCTCCAGGCAGCCGTCACCG |
| Sequence-based reagent | L42C_F | This paper | PCR primers | CAGCGGTGACGGCAAGTGCGAGGCCCT |
| Sequence-based reagent | L42C_R | This paper | PCR primers | GACGTAGAGGGCCTCGCACTTGCCGTCA |
| Sequence-based reagent | E43C_F | This paper | PCR primers | GCGGTGACGGCAAGCTGTGCGCCCTCTA |
| Sequence-based reagent | E43C_R | This paper | PCR primers | GGACGTAGAGGGCGCACAGCTTGCCGTC |

*Appendix 1 Continued on next page*

*Appendix 1 Continued*

| Reagent type (species) or resource | Designation | Source or reference | Identifiers | Additional information |
|---|---|---|---|---|
| Sequence-based reagent | A44C_F | This paper | PCR primers | CGGCAAGCTGGAGTGCCTCTACGTCCTC |
| Sequence-based reagent | A44C_R | This paper | PCR primers | GAGGACGTAGAGGCACTCCAGCTTGCCG |
| Sequence-based reagent | Y46A_F | This paper | PCR primers | GGAGGCCCTCTGCGTCCTCATGGTAC |
| Sequence-based reagent | Y46A_R | This paper | PCR primers | GTACCATGAGGACGGCGAGGGCCTCC |
| Sequence-based reagent | W323A_F | This paper | PCR primers | GGTCTTCCCGACCGCCGTCTGGGGCAC |
| Sequence-based reagent | W323A_R | This paper | PCR primers | GTGCCCCAGACGGCGGTCGGGAAGACC |
| Sequence-based reagent | W323C_F | This paper | PCR primers | GGTCTTCCCGACACACGTCTGGGGCAC |
| Sequence-based reagent | W323C_R | This paper | PCR primers | GTGCCCCAGACGTGTGTCGGGAAGACC |
| Sequence-based reagent | V324A_F | This paper | PCR primers | CCCCAGACGTGGGCCGGGAAGACCATC |
| Sequence-based reagent | V324A_R | This paper | PCR primers | GATGGTCTTCCCGGCCCACGTCTGGGG |
| Sequence-based reagent | V324W_F | This paper | PCR primers | CCCCAGACGTGGTGGGGGAAGACCATC |
| Sequence-based reagent | V324W_R | This paper | PCR primers | GATGGTCTTCCCCCACCACGTCTGGGG |
| Sequence-based reagent | Q147C_F | This paper | PCR primers | CAGGGCGGCATAGCACTCGATGGTGGAC |
| Sequence-based reagent | Q147C_R | This paper | PCR primers | GTCCACCATCGAGTGCTATGCCGCCCTG |
| Sequence-based reagent | Y148C_F | This paper | PCR primers | GCCAGGGCGGCACACTGCTCGATGGTG |
| Sequence-based reagent | Y148C_R | This paper | PCR primers | CACCATCGAGCAGTGTGCCGCCCTGGC |
| Peptide, recombinant protein | eGFP in pcDNA3 | Gift | | Enhanced Green Fluorescent Protein |

*Appendix 1 Continued on next page*

*Appendix 1 Continued*

| Reagent type (species) or resource | Designation | Source or reference | Identifiers | Additional information |
|---|---|---|---|---|
| Commercial assay or kit | Midiprep | ThermoFisher Scientific | Cat# K210004 | DNA extraction kit |
| Chemical compound, drug | DIDS | Sigma-Aldrich | CAS # 53005-05-3 | 4,4'-diisothiocyano-2,2'-stilbenedisulfonic acid (stock 50 mM) |
| Chemical compound, drug | Mef | Sigma-Aldrich | CAS # 61-68-7 | mefenamic Acid (stock 50 mM) |
| Software, algorithm | pCLAMP 11 | Molecular Devices | | pCLAMP 11 software |
| Software, algorithm | GraphPad Prism 9 | GraphPad Software | | GraphPad Prism 9 software |
| Software, algorithm | ICM-pro | MolSoft LLC | | ICM-pro 3.8 software |
| Software, algorithm | GROMACS | Royal Institute of Technology and Uppsala University, Sweden | | GROMACS 2021.4 |
| Software, algorithm | TTClust | Thibault Tubiana, PhD | | TTClust, a molecular simulation clustering program |
| Other | Axopatch 200B amplifier | Molecular Devices | | Axopatch 200B amplifier |
| Other | Digidata 1440 A digitizer | Molecular Devices | | Digidata 1440 A digitizer |
| Other | Lipofectamine 2000 | ThermoFisher Scientific | Cat # 11668019 | Lipofectamine 2000 transfection reagent |

