## [Editor Report · eLife assessment]

By combining electrophysiological analysis of mutant channels and molecular dynamics simulations, this **important** study identifies a common binding site for two structurally distinct activators of KCNQ1-KCNE1 channels. The findings represent an **important** advance for the field, with **convincing** functional and computational data to support the claims. The work will be of interest to those studying the binding of small molecule drugs to membrane protein complexes.

---

## [Referee Report · Reviewer #1 (Public Review)]

Chan et al. attempted to identify the binding sites or pockets for the KCNQ1-KCNE1 activator mefenamic acid. Because the KCNQ1-KCNE1 channel is responsible for cardiac repolarization, genetic impairment of either the KCNQ1 or KCNE1 gene can cause cardiac arrhythmias. Therefore, the development of activators without side effects is highly desired. Since mefenamic acid binding requires both KCNQ1 and KCNE1 subunits, the authors performed drug docking simulations using the KCNQ1-psKCNE1 structural model with substitution of the extracellular five amino acids (R53-Y58) of KCNE3 to D39-A44 of KCNE1. They successfully identified some critical amino acid residues, including W323 of KCNQ1 and K41 and A44 of KCNE1. They then tested these identified amino acid residues by analyzing the point mutants and confirmed that they were critical for the binding of the activator. They also examined another activator, but structurally different DIDS, and reported that DIDS and mefenamic acid share the binding pocket, and concluded that the extracellular region composed of S1, S6, and KCNE1 is a generic binding pocket for the IKS activators.

The limitation of this study is that they had to use the KCNQ1-KCNE3-based structural model for the docking simulation. Although they only focused on the extracellular region substituted by the six amino acid residues of KCNE1, the binding mode or location of KCNE1 might be different from KCNE3. Another weakness is that unbinding may be facilitated in the closed state, whereas they had to use the open channel for the MD simulation. Therefore, their MD simulations do not necessarily reflect the unbinding process in the closed state, which should occur in the comparable electrophysiological experiments. Nevertheless, the data are solid and well support their conclusions. This work should be valuable to the field, not only for future drug design but also for the biophysical understanding of the binding/unbinding of drugs to ion channel complexes.

---

## [Referee Report · Reviewer #2 (Public Review)]

The voltage-gated potassium channel KCNQ1/KCNE1 (IKs) plays important physiological functions, for instance in the repolarization phase of the cardiac action potential. Loss-of-function of KCNQ1/KCNE1 is linked to disease. Hence, KCNQ1/KCNE1 is a highlighted pharmacological target and mechanistic insights into how channel modulators enhance the function of the channel is of great interest. The authors have through several previous studies provided mechanistic insights into how small-molecule activators like ML277 act on KCNQ1. However, less is known about the binding site and mechanism of action of other type of channel activators, which require KCNE1 for their effect. In this study, Chan and co-workers use molecular dynamics approaches, mutagenesis and electrophysiology to propose an overall similar binding site for the KCNQ1/KCNE1 activators mefenamic acid and DIDS, located at the extracellular interface of KCNQ1 and KCNE1. The authors propose an induced-fit model for the binding site, which critically engages residues in the N-terminus of KCNE1. Moreover, the authors discuss possible mechanisms of action of how drug binding to this site may enhance channel function.

The authors address an important question, of broad relevance to researchers in the field. The manuscript is well written and the text easy to follow. A strength of the work is the parallel use of experimental and simulation approaches, which enables both functional testing and mechanistic predictions and interpretations. For instance, the authors have experimentally assessed the putative relevance of a large set of residues based on simulation predictions. A minor limitation is that not all residues of putative importance for drug binding/effects can be reliable evaluated in experiments, which is, however, clearly discussed by the authors and a challenge shared by electrophysiologists in the field.

---

## [Referee Report · Reviewer #3 (Public Review)]

The authors identified the mefenamic (Mef) binding site and DIDS binding site on the KCNQ1 KCNE1 complex. The authors also identified the mechanism of interactions using electrophysiological recording, calculating V1/2 of different mutants, and looking at the instantaneous and tail currents. The contribution of each residue within the binding pocket was analysed using GBSA and PBSA and traditional molecular dynamics simulation.

The manuscript has been substantially revised from the previous version with a greater depth of computational analysis.

---

## [Author Response]

The following is the authors’ response to the original reviews.

**Reviewer #1 (Public Review):**

Chan et al. tried identifying the binding sites or pockets for the KCNQ1-KCNE1 activator mefenamic acid. Because the KCNQ1-KCNE1 channel is responsible for cardiac repolarization, genetic impairment of either the KCNQ1 or KCNE1 gene can cause cardiac arrhythmia. Therefore, the development of activators without side effects is highly demanded. Because the binding of mefenamic acid requires both KCNQ1 and KCNE1 subunits, the authors performed drug docking simulation by using KCNQ1-KCNE3 structural model (because this is the only available KCNQ1-KCNE structure) with substitution of the extracellular five amino acids (R53-Y58) into D39-A44 of KCNE1. That could be a limitation of the work because the binding mode of KCNE1 might differ from that of KCNE3. Still, they successfully identified some critical amino acid residues, including W323 of KCNQ1 and K41 and A44 of KCNE1. They subsequently tested these identified amino acid residues by analyzing the point mutants and confirmed that they attenuated the effects of the activator. They also examined another activator, yet structurally different DIDS, and reported that DIDS and mefenamic acid share the binding pocket, and they concluded that the extracellular region composed of S1, S6, and KCNE1 is a generic binding pocket for the IKS activators.The data are solid and well support their conclusions, although there are a few concerns regarding the choice of mutants for analysis and data presentation.Other comments:1. One of the limitations of this work is that they used psKCNE1 (mostly KCNE3), not real KCNE1, as written above. It is also noted that KCNQ1-KCNE3 is in the open state. Unbinding may be facilitated in the closed state, although evaluating that in the current work is difficult.

We agree that it is difficult to evaluate the role of unbinding from our model. Our data showing that longer interpulse intervals have a normalizing effect on the GV curve (Figure 3-figure supplement 2) could be interpreted to suggest that unbinding occurs in the closed state. Alternatively, the slowing of deactivation caused by S1-S6 interactions and facilitated by the activators may effectively be exceeded at the longer interpulse intervals.

1. According to Figure 2-figure supplement 2, some amino acid residues (S298 and A300) of the turret might be involved in the binding of mefenamic acid. On the other hand, Q147 showing a comparable delta G value to S298 and A300 was picked for mutant analysis. What are the criteria for the following electrophysiological study?

EP experiments interrogated selected residues with significant contributions to mefenamic acid and DIDs coordination as revealed by the MM/GBSA and MM/PBSA methods. A300 was identified as potentially important. We did attempt A300C but were never able to get adequate expression for analysis.

1. It is an interesting speculation that K41C and W323A stabilize the extracellular region of KCNE1 and might increase the binding efficacy of mefenamic acid. Is it also the case for DIDS? K41 may not be critical for DIDS, however.

Yes, we found K41 was not critical to the binding/action of DIDS compared to MEF. In electrophysiological experiments with the K41C mutation, DIDS induced a leftward GV shift (~ -25 mV) whereas the normalized response was statistically non-significant. In MD simulation studies, we observed detachment of DIDS from K41C-Iks only in 3 runs out of 8 simulations. This is in contrast to Mef, where the drug left the binding site of K41C-Iks complex in all simulations.

1. Same to #2, why was the pore turret (S298-A300) not examined in Figure 7?

Again, we attempted A300C but could not get high enough expression.

**Reviewer #3 (Public Review):**

Weaknesses:1. The computational aspect of the work is rather under-sampled - Figure 2 and Figure 4. The lack of quantitative analysis on the molecular dynamic simulation studies is striking, as only a video of a single representative replica is being shown per mutant/drug. Given that the simulations shown in the video are extremely short; some video only lasts up to 80 ns. Could the author provide longer simulations in each simulation condition (at least to 500 ns or until a stable binding pose is obtained in case the ligand does not leave the binding site), at least with three replicates per each condition? If not able to extend the length of the simulations due to resources issue, then further quantitative analysis should be conducted to prove that all simulations are converged and are sufficient. Please see the rest of the quantitative analysis in other comments.

We provide more quantitative analysis for the existing MD simulations and ran five additional simulations with 500 ns duration by embedding the channel in a POPC lipid membrane. For the new MD simulations, we used a different force field in order to minimize ambiguity related to force fields as well. Analysis of these data has led to new data and supplemental figures regarding RMSD of ligands during the simulations (Figure 4-figure supplement 1 and Figure 6-figure supplement 3), clustering of MD trajectories based on Mef conformation (Figure 2-figure supplement 3 and Figure 6 -figure supplement 2), H-bond formation over the simulations (Figure 2-figure supplement 4 and Figure 6-figure supplement 1). We have edited the manuscript to include this new information where appropriate.

1. Given that the protein is a tetramer, at least 12 datasets could have been curated to improve the statistic. It was also unclear how frequently the frames from the simulations were taken in order to calculate the PBSA/GBSA.

By using one ligand for each ps-IKs channel complex we tried to keep the molecular system and corresponding analysis as simple as was possible. Our initial results have shown that 4D docking and subsequent MD simulations with only one ligand bound to ps-IKs was complicated enough. Our attempts to dock 4 ligands simultaneously and analyze the properties of such a system were ineffective due to difficulties in: (i) obtaining stable complexes during conformational sampling and 4D docking procedures, since the ligand interaction covers a region including three protein chains with dynamic properties, (ii) possible changes of receptor conformation properties at three other subunits when one ligand is already occupying its site, (iii) marked diversity of the binding poses of the ligand as cluster analysis of ligand-channels complex shows (Figure 2-figure supplement 3).We have added a line in the methods to clarify the use of only one ligand per channel complex in simulations.

In order to calculate MMPBSA/MMGBSA we used a frame every 0.3 ns throughout the 300 ns simulation (1000 frames/simulation) or during the time the ligand remained bound. We have clarified this in the Methods.

1. The lack of labels on several structures is rather unhelpful (Figure 2B, 2C, 4B). The lack of clarity of the interaction map in Figures 2D and 6A.

We updated figures considering the reviewer's comments and added labels. For 2D interaction maps, we provided additional information in figure legends to improve clarity.

1. The RMSF analysis is rather unclear and unlabelled thoroughly. In fact, I still don't quite understand why n = 3, given that the protein is a tetramer. If only one out of four were docked and studied, this rationale needs to be explained and accounted for in the manuscript.

The rationale of conducting MD simulations with one ligand bound to IKs is explained in response to point 2 of the reviewer’s comments.

RMSF analysis in Figure 4C-E was calculated using the chain to which Mef was docked but after Mef had left the binding site. Details were added to the methods.

1. For the condition that the ligands suppose to leave the site (K42C for Mef and Y46A for DIDS), can you please provide simulations at a sufficient length of time to show that ligand left the site over three replicates? Given that the protein is a tetramer, I would be expecting three replicates of data to have four data points from each subunit. I would be expecting distance calculation or RMSD of the ligand position in the binding site to be calculated either as a time series or as a distribution plot to show the difference between each mutant in the ligand stability within the binding pocket. I would expect all the videos to be translatable to certain quantitative measures.

We have shown in the manuscript that the MEF molecule detaches from the K41C/IKs channel complex in all three simulations (at 25 ns, 70 ns and 20 ns, Table. 4). Similarly, the ligand left the site in all five new 500 ns duration simulations. We did not provide simualtions for Y46A, but Y46C left the binding site in 4 of 5 500 ns simulations and changed binding pose in the other.

Difficulties encountered upon extending the docking and MD simulations for 4 receptor sites of the channel complex is discussed in our response to point # 2 of the reviewer.

1. Given that K41 (Mef) and Y46 are very important in the coordination, could you calculate the frequency at which such residues form hydrogen bonds with the drug in the binding site? Can you also calculate the occupancy or the frequency of contact that the residues are making to the ligand (close 4-angstrom proximity etc.) and show whether those agree with the ligand interaction map obtained from ICM pro in Figure 2D?

We thank the reviewer for the suggestion to analyze the H-bond contribution to ligand dynamics in the binding site. In the plots shown in Figure 2-figure supplement 4 and Figure 6-figure supplement 1, we now provide detailed information about the dynamics of the H-bond formation between the ligand and the channel-complex throughout simulations. In addition, we have quantified this and have added these numbers to a table (Table 2) and in the text of the results.

1. Given that the author claims that both molecules share the same binding site and the mode of ligand binding seems to be very dynamic, I would expect the authors to show the distribution of the position of ligand, or space, or volume occupied by the ligand throughout multiple repeats of simulations, over sufficient sampling time that both ligand samples the same conformational space in the binding pocket. This will prove the point in the discussion - Line 463-464. "We can imagine a dynamic complex... bind/unbind from Its at a high frequency".

To support our statement regarding a dynamic complex we analyzed longer MD simulations and clustered trajectories, from this an average conformation from each cluster was extracted and provided as supplementary information which shows the different binding modes for Mef (Figure 2-figure supplement 3). DIDS was more stable in MD simulations and though there were also several clusters, they were similar enough that when using the same cut-off distance as for mefenamic acid, they could be grouped into one cluster. (Note the scale differences on dendrogram between Figure 2-figure supplement 3 and Figure 6-figure supplement 2).

1. I would expect the authors to explain the significance and the importance of the PBSA/GBSA analysis as they are not reporting the same energy in several cases, especially K41 in Figure 2 - figure supplement 2. It was also questionable that Y46, which seems to have high binding energy, show no difference in the EPhys works in figure 3. These need to be commented on.

Several studies indicate that ΔG values calculated using MM/PBSA and MM/GBSA methods may vary. Some studies report marked differences and the reasons for such a discrepancy is thoroughly discussed in a review by Genheden and Ryde (PMID: 25835573). Therefore, we used both methods to be sure that key residues contributing to ligand binding identified with one method appear in the list of residues for which the calculations are done with the other method.

Y46C which showed only a slightly less favorable binding energy and did not unbind during 300 ns simulations, unbound, or changed pose in 4 out of 5 of the longer simulations in the presence of a lipid membrane (Figure 4-figure supplement 1). The discrepancy between electrophysiological and MD data is commented in the manuscript (pages 12-13).

1. Can the author prove that the PBSA/GBSA analysis yielded the same average free energy throughout the MD simulation? This should be the case when the simulations are converged. The author may takes the snapshots from the first ten ns, conduct the analysis and take the average, then 50, then 100, then 250 and 500 ns. The author then hopefully expects that as the simulations get longer, the system has reached equilibrium, and the free energy obtained per residue corresponds to the ensemble average.

As we mention in the manuscript, MEF- channel interactions are quite dynamic and vary even from simulation to simulation. The frequent change of the binding pose of the ligands observed during simulations (represented in Figure 2 - figure supplement 3 as clusters) is a clear reflection of such a dynamic process. Therefore, we do not expect the same average energy throughout the simulation but we do expect that ΔG values stands above the background for key residues, which was generally the case (Figure 2 - figure supplement 2 and Figure 6.)

1. The phrase "Lowest interaction free energy for residues in ps-KCNE1 and selected KCNQ1 domains are shown as enlarged panels (n=3 for each point)" needs further explanation. Is this from different frames? I would rather see this PBSA and GBSA calculated on every frame of the simulations, maybe at the one ns increment across 500 ns simulations, in 4 binding sites, in 3 replicas, and these are being plotted as the distribution instead of plotting the smallest number. Can you show each data point corresponding to n = 3?

The MMPBSA/MMGBSA was calculated for 1000 frames across 3x300 ns simulations with 0.3 ns sampling interval, together 3000 frames, shown in Figure 2-figure supplement 2 and includes error bars to show the differences across runs. We have updated the legend for greater clarity.

1. I cannot wrap my head around what you are trying to show in Figure 2B. This could be genuinely improved with better labelling. Can you explain whether this predicted binding pose for Mef in the figure is taken from the docking or from the last frame of the simulation? Given that the binding mode seems to be quite dynamic, a single snapshot might not be very helpful. I suggest a figure describing different modes of binding. Figure 2B should be combined with figure 2C as both are not very informative.

We have updated Figure 2B with better labelling and added a new figure showing the different modes of binding (Figure 2-figure supplement 3).

1. Similar to the comment above, but for Figure 4B. I do not understand the argument. If the author is trying to say that the pocket is closed after Mef is removed - then can you show, using MD simulation, that the pocket is openable in an apo to the state where Mef can bind? I am aware that the open pocket is generated through batches of structures through conformational sampling - but as the region is supposed to be disordered, can you show that there is a possibility of the allosteric or cryptic pocket being opened in the simulations? If not, can you show that the structure with the open pocket, when the ligand is removed, is capable of collapsing down to the structure similar to the cryo-EM structure? If none of the above work, the author might consider using PocketMiner tools to find an allosteric pocket (https://doi.org/10.1038/s41467-023-36699-3) and see a possibility that the pocket exists.

Please see Figure 4 – figure supplement 2 which depicts the binding pocket from the longest run we performed (1250 ns) before drug detachment (grey superimposed structures) and after (red superimposed structures). Mefenamic acid is represented as licorice and colored green. Snapshots for superimposition were collected every 10 ns. As can be seen in the figure, when the drug leaves the binding site (after 500 ns, structures colored red), the N-terminal residue of psKCNE1, W323, and other residues that form the pocket shift toward the binding site, overlapping with where Mefenamic acid once resided. The surface structure in Figure 4B shows this collapse.

In the manuscript, we propose that drug binding occurs by the mechanism that could be best described by induced fit models, which state that the formation of the firm complexes (channel-Mef complex) is a result of multiple-states conformational adjustments of the bimolecular interaction. These interactions do not necessarily need to have large interfaces at the initial phase. This seems to be the case in Mef with IKS interactions, since we could not identify a pocket of appropriate size either using PocketMiner software suggested by the reviewer or with PocketFinder tool of ICM-pro software.

1. Figure 4C - again, can you show the RMSF analysis of all four subunits leading to 12 data points? If it is too messy to plot, can you plot a mean with a standard deviation? I would say that a 1-1.5 angstroms increase in the RMSF is not a "markedly increased", as stated on line 280. I would also encourage the authors to label whether the RMSF is calculated from the backbone, side-chain or C-alpha atoms and, ideally, compare them to see where the dynamical properties are coming from.

Please see the answer to comment #4. We agree that the changes are not so dramatic and modified the text accordingly. RMSD was calculated for backbone atom to compare residues with different side chains, a note of this is now in the methods and statistical significance of ps-IKs vs K41C, W323A and Y46C is indicated in Figures 4C-4E.

1. In the discussion - Lines 464-467. "Slowed deactivation of the S1/KCNE1/Pore domain/drug complex... By stabilising the activated complex. MD simulation suggests the latter is most likely the case." Can you point out explicitly where this has been proven? If the drug really stabilised the activated complex, can you show which intermolecular interaction within E1/S1/Pore has the drug broken and re-form to strengthen the complex formation? The authors have not disproven the point on steric hindrance either. Can this be disproved by further quantitative analysis of existing unbiased equilibrium simulations?

The stabilization of S1/KCNE1/Pore by drugs does not necessarily have to involve a creation of new contacts between protein parts or breakage of interfaces between them. The stabilization of activated complexes by drugs may occur when the drug simultaneously binds to both moveable parts of the channel, such as voltage sensor(s) or upper KCNE1 region, and static region(s) of the channel, such as the pore domain. We have changed the corresponding text for better clarity.

1. Figure 4D - Can you show this RMSF analysis for all mutants you conducted in this study, such as Y46C? Can you explain the difference in F dynamics in the KCNE3 for both Figure 4C and 4D?

We now show the RMSF for K41C, W323A and Y46C in Figure 4C-E. We speculate that K41 (magenta) and W323 (yellow), given their location at the lipid interface (see Author response image 1), may be important stabilizing residues for the KCNE N-terminus, whereas Y46 (green) which is further down the TMD has less of an impact.

**Author response image 1. sa4fig1:** 

1. Line 477: the author suggested that K41 and Mef may stabilise the protein-protein interface at the external region of the channel complex. Can you prove that through the change in protein-protein interaction, contact is made over time on the existing MD trajectories, whether they are broken or formed? The interface from which residues help to form and stabilise the contact? If this is just a hypothesis for future study, then this has to be stated clearly.

It is known that crosslinking of several residues of external E1 with the external pore residues dramatically stabilizes voltage-sensors of KCNQ1/KCNE1 complex in the up-state conformation. This prevents movable protein regions in the voltage-sensors returning to their initial positions upon depolarization, locking the channel in an open state. We suggest that MEF may restrain the backward movement of voltage-sensors in a similar way that stabilizes open conformation of the channel. The stabilization of the voltage sensor domain through MEF occurs due to contacts of the drug with both static (pore domain) and dynamic protein parts (voltage-sensors and external KCNE1 regions). We have changed the corresponding part of the text.

1. The author stated on lines 305-307 that "DIDS is stabilised by its hydrophobic and vdW contacts with KCNQ1 and KCNE1 subunits as well as by two hydrogen bonds formed between the drug and ps-KCNE1 residue L42 and KCNQ1 residue Q147" Can you show, using H-bond analysis that these two hydrogen bonds really exist stably in the simulations? Can you show, using minimum distance analysis, that L42 are in the vdW radii stably and are making close contact throughout the simulations?

We performed a detailed H-bond analysis (Figure 6-supplement figure 1) which shows that DIDS forms multiple H-bond over the simulations, though only some of them (GLU43, TYR46, ILE47, SER298, TYR299, TRP323 ) are stable. Thus, the H-bonds that we observed in DIDS-docking experiments were unstable in MD simulations. As in the case of the IKs-MEF complex, the prevailing H-bonds exhibit marked quantitative variability from simulation to simulation. We have added a table detailing the most frequent H-bonds during MD simulations (Table 2).

1. Discussion - In line 417, the author stated that the "S1 appears to pull away from the pore" and supplemented the claim with the movie. This is insufficient. The author should demonstrate distance calculation between the S1 helix and the pore, in WT and mutants, with and without the drug. This could be shown as a time series or distribution of centre-of-mass distance over time.

We tried to analyze the distance changes between the upper S1 and the pore domain but failed to see a strong correlation We have removed this statement from the discussion.

1. Given that all the work were done in the open state channel with PIP2 bound (PDB entry: 6v01), could the author demonstrate, either using docking, or simulations, or alignment, or space-filling models - that the ligand, both DIDS and Mef, would not be able to fit in the binding site of a closed state channel (PDB entry: 6v00). This would help illustrate the point denoted Lines 464-467. "Slowed deactivation of the S1/KCNE1/Pore domain/drug complex... By stabilising the activated complex. MD simulation suggests the latter is most likely the case."

As of now, a structure representing the closed state of the channel does not exist. 6V00 is the closed inactivated state of the channel pore with voltage-sensors in the activated conformation. In order to create simulation conditions that reliably describe the electrophysiological experiments, at least a good model for closed channels with resting state voltage sensors is necessary.

1. The author stated that the binding pose changed in one run (lines 317 to 318). Can you comment on those changes? If the pose has changed - what has it changed to? Can you run longer simulations to see if it can reverse back to the initial confirmation? Or will it leave the site completely?

Longer simulations and trajectory clustering revealed several binding modes, where one pose dominated in approximately 50% of all simulations in Figure 2-figure supplement 3 encircled with a blue frame.

1. Binding free energy of -32 kcal/mol = -134 kJ/mol. If you try to do dG = -RTlnKd, your lnKd is -52. Your Kd is e^-52, which means it will never unbind if it exists. I am aware that this is the caveat with the methodologies. But maybe these should be highlighted throughout the manuscript.

We thank the reviewer for this comment. ΔΔG values, and corresponding Kd values, calculated from simulation of Mef-ps-IKs complex do not reflect the apparent Kd values determined in electrophysiological experiments, nor do they reflect Kd values of drug binding that could be determined in biochemical essays. Important measures are the changes observed in simulations of mutant channel complexes relative to wild type. We now briefly mention this issue in the manuscript.

**Reviewer #1 (Recommendations For The Authors):**
1. It would be nice to have labels of amino acid residues in Figure 2B.

We updated Figure 2B and added some residue labels.

1. Fig. 3A and 7A. In what order the current traces are presented? I don't see the rule.

We have now arranged the current traces in a more orderly manner, listing them first by ascending KCNE1 residue numbers and then by ascending KCNQ1 residue numbers. Now consistent with Fig 3 and 7 (normalized response and delta V1/2).

1. Line 312 "A44 and Y46 were more so." A44 may be more critical, but I can't see Y46 is more, according to Figure 2-figure supplement2 and Figure 6.

Indeed, comparison of the energy decomposition data indicates approximately the same ΔG values for Y46. We have revised this in the text correspondingly.

1. Line 267 "Mefenamic acid..." I would like to see the movie.

We no longer have access to this original movie

1. In supplemental movies 5-7, the side chains of some critical amino acid residues (W323, K41) would be better presented as in movies 1-4.

We have retained the original presentations of these movies as the original files are no longer available.

**Reviewer #2 (Recommendations For The Authors):**
General comments:1. To determine the effect of mefenamic acid and DIDS on channel closing kinetics, a protocol in which they step from an activating test pulse to a repolarizing tail pulse to -40 mV for 1 s is used. If I understand it right, the drug response is assessed as the difference in instantaneous tail current amplitude and the amplitude after 1 s (row 599-603). The drug response of each mutant is then normalized to the response of the WT channel. However, for several mutants there is barely any sign of current decay during this relatively brief pulse (1 s) at this specific voltage. To determine drug effects more reliably on channel closing kinetics/the extent of channel closing, I wonder if these protocols could be refined? For instance, to cover a larger set of voltages and consider longer timescales?

To clarify, the drug response of each mutant is not normalized to the response of the WT channel. In fact, our analysis is not meant to compare mutant and WT tail current decay but rather how isochronal tail current decay is changed in response to drug treatment in each channel construct. As acknowledged by the reviewer, the peak to end difference currents were calculated by subtracting the minimum amplitude of the deactivating current from the peak amplitude of the deactivating current. But the difference current in mefenamic acid or DIDS was normalized to the maximum control (in the absence of drug) difference current and subtracted from 1.0 to obtain the normalized response. Thus, the difference in tail current decay in the absence and in the presence of drug is measured within the same time scale and allow a direct comparison between before and after drug treatment. As shown in Fig 3D and 7C, a large drug response such as the one measured in WT channels is reflected by a value close to 1. A smaller drug response is indicated by low values. We recognize that some mutations resulted in an intrinsic inhibition of tail current decay in the absence of drug, which potentially lead to underestimating the normalized response value.Our goal was not to study in detail the effects of the drug on channel closing kinetics, but only to determine the impact of the mutation on drug binding by using tail current decay as a readout. Consequently, we believe that the duration of the deactivating tail current used in this experiment was sufficient to detect drug-induced tail current decay inhibition.

1. The effect of mefenamic acid seems to be highly dependent on the pulse-to-pulse interval in the experiments. For instance, for WT in Figure 3 - Figure supplement 1, a 15 s pulse-to-pulse interval provides a -100 mV shift in V1/2 induced by mefenamic acid, whereas there is no shift induced when using a 30 s pulse-to-pulse interval. Can the authors explain why they generally consider a 15 s pulse-to-pulse interval more suitable (physiologically relevant?) in their experiments to assess drug effects?

In our previous experiments, we have determined that a 15 s inter-pulse interval is generally adequate for the WT IKs channels to fully deactivate before the onset of the next pulse. Consistent with our previous work (Wang et al. 2019), we observed that in wild-type EQ channels, there is no current summation from one pulse to the next one (see Fig 1A, bottom panel). This is important as the IKs channel complex is known to be frequency dependent i.e. current amplitude increases as the inter-pulse interval gets shorter. Such current summation results in a leftward shift of the conductance-voltage (GV) relationship. This is also important with regards to drug effects. As indicated by the reviewer, mefenamic acid effects are prominent with a 15 sec inter-pulse interval but less so with a 30 sec inter-pulse interval when enough time is given for channels to more completely deactivate. Full effects of mefenamic acid would have therefore been concealed with a 30sec inter-pulse interval.

Moreover, our patch-clamp recordings aim to explore the distinct responses of mutant channels to mefenamic acid and DIDS in comparison to the wild-type channel. It is important to note that the inter-pulse interval's physiological relevance is not necessarily crucial in this context.

1. Related to comment 1 and 2, there is a large diversity in the intrinsic properties of tested mutants. For instance, V1/2 ranges from 4 to 70 mV. Also, there is large variability in the slope of the G-V curves. Whether channel closing kinetics, or the impact of pulse-to-pulse interval, vary among mutants is not clear. Could the authors please discuss whether the intrinsic properties of mutants may affect their ability to respond to mefenamic acid and DIDS? Also, please provide representative current families and G-V curves for all assessed mutants in supplementary figures.

The intrinsic properties of some mutants vary from the WT channels and influence their responsiveness to mefenamic acid and DIDS. The impact of the mutations on the IKs channel complex are reflected by changes in V1/2 (Table 1, 4) and tail current decay (Figs. 3, 7). But, it is the examination of the drug effects on these intrinsic properties (i.e. GV curve and tail current decay) that constitutes the primary endpoint of our study. We consider that the degree by which mef and DIDS modify these intrinsic properties reflects their ability to bind or not to the mutated channel. In our analysis, we compared each mutant's response to mefenamic acid and DIDS with its respective control. Consequently, the intrinsic properties of the mutant channels have already been considered in our evaluation. As requested, we have provided representative current families and G-V curves for all assessed mutants in Figure 3-figure supplement 1 and Figure 7-figure supplement 1.

1. The A44C and Y148C mutants give strikingly different currents in the examples shown in Figure 3 and Figure 7. What is the reason for this? In the examples in figure 7, it almost looks like KCNE1 is absent. Although linked constructs are used, is there any indication that KCNE1 is not co-assembled properly with KCNQ1 in those examples?

The size of the current is critical to determining its shape, as during the test pulse there is some endogenous current mixed in which impacts shape. A44C and Y148C currents shown in Figure 7 are smaller with a larger contribution of the endogenous current, mostly at the foot of the current trace. In our experience there is little endogenous current in the tail current at -40 mV and for this reason we focus our measurements there.

Although constructs with tethered KCNQ1 and KCNE1 were used, we cannot rule out the possibility that Q1 and E1 interaction was altered by some of the mutations. Several KCNE1 and KCNQ1 residues have been identified as points of contact between the two subunits. For instance, the KCNE1 loop (position 36-47) has been shown to interact with the KCNQ1 S1-S2 linker (position 140-148) (Wang et al, 2011). Thus, it is conceivable that mutation of one or several of those residues may alter KCNQ1/KCNE1 interaction and modify the activation/deactivation kinetics of the IKs channel complex.

1. I had a hard time following the details of the simulation approaches used. If not already stated (I could not find it), please provide: (i) details on whether the whole channel protein was considered for 4D docking or a docking box was specified, (ii) information on how simulations with mutant ps-IKs were prepared (for instance with the K41C mutant), especially whether the in silico mutated channel was allowed to relax before evaluation (and for how long). Also, please make sure that information on simulation time and number of repeats are provided in the Methods section.

For 4D docking, only residues within 0.8 nm of psKCNE1 residues D39-A44 were selected. Complexes with mutated residues were relaxed using the same protocol as the WT channel, (equilibration with gradually releasing restraints with a final equilibration for 10 ns where only the backbone was constrained with 50 kcal/mol/nm2). We have updated the methods accordingly.

Specific comments:In figure legends, please provide information on whether data represents mean +/- SD or SEM. Also, please provide information on which statistical test was used in each figure.

We revised the figure legend to add the nature of the statistical test used.

G-V curves are normalized between 0 and 1. However, for many mutants the G-V relationship does not reach saturation at depolarized voltages. Does this affect the estimated V1/2? I could not really tell as I was not sure how V1/2 was determined for different mutants (could the explanation on row 595-598 be clarified)?

The primary focus here is in the shift between the control response and drug response for each mutant, rather than the absolute V1/2 values. The isochronal G-V curves that are generated for each construct (WT and mutant) utilize an identical voltage protocol. This approach ensures a uniform comparison among all mutants. By observing the shifts in these curves, we can gain insight into the response of mutant channels to the drug. This information ultimately helps elucidate the inherent properties of the mutant channels and contributes to our understanding of the drug's binding mechanism to the channel.

As requested by the reviewer, we also clarified the way V1/2 was generated: When the G-V curve did not reach zero, the V1/2 value was directly read from the plot at the voltage point where the curve crossed the 0.5 value on the y coordinate.

A general comment is that the Discussion is fairly long and some sections are quite redundant to the Results section. The authors could consider focusing the text in the Discussion.

We changed the discussion correspondingly wherever it was appropriate.

I found it a bit hard to follow the authors interpretation on whether their drug molecules remain bound throughout the experiments, or whether there is fast binding/unbinding. Please clarify if possible.

In the 300 ns MD simulations mefenamic acid and DIDS remained stably bound to WT-ps-IKS, binding of drugs to mutant complexes are described in the Table 3 and Table 5. In longer simulations with the channel embedded in a lipid environment, mefenamic acid unbinds in two out of five runs for WT-ps-IKs (Figure 4 – figure supplement 1), and DIDS shows a few events where it briefly unbinds (Figure 6 -figure supplement 3). Based on electrophysiological data we speculate that drugs might bind and unbind to WT-ps-IKs during the gating process. We do not see bind-unbinding in MD simulations, since the model we used in simulations reflects only open conformation of the channel-complex with an activated-state voltage-sensor, whereas a resting-state voltage sensor condition was not considered.

The authors have previously shown that channels with no, one or two KCNE1 subunits are not, or only to a small extent, affected by mefenamic acid (Wang et al., 2020). Could the details of the binding site and proposed mechanisms of action provide clues as to why all binding sites need to be occupied to give prominent drug effects?

In the manuscript, we propose that the binding of drugs induces conformational changes in the pocket region that stabilize S1/KCNE1/Pore complex. In the tetrameric channel with 4:4 alpha to beta stoichiometry the drugs are likely to occupy all four sites with complete stabilization of S1/KCNE1/Pore. When one or more KCNE1 subunits is absent, as in case of EQQ, or EQQQQ constructs, drugs will bind to the site(s) where KCNE1 is available. This will lead to stabilization of the only certain part of the S1/KCNE1/Pore complex. We believe that the corresponding effect of the drug, in this case will be partially effective.

There is a bit of jumping in the order of when some figures are introduced (e.g. row 178 and 239). The authors could consider changing the order to make the figures easier to follow.

We have changed the corresponding section appropriately to improve the reading flow.

Row 237: "Data not shown", please show data.

The G-V curve of the KCNE1 Y46C mutant displays a complex, double Boltzmann relationship which does not allow for the calculation of a meaningful V1/2 nor would it allow for an accurate determination of drug effects. Consequently, we have excluded it from the manuscript.

In the Discussion, the author use the term "KCNE1/3". Does this correspond to the previous mention of "ps-KCNE1"?

Yes, this refers to ps-KCNE1. We have changed it correspondingly.

Row 576: When was HMR 1556 used?

While HMR 1556 was used in preliminary experiments to confirm that the recorded current was indeed IKs, it does not provide substantial value to the data presented in our study or our experiments. As a result, we have excluded HMR 1556 experiments from the final results and have revised the Methods section accordingly.

**Reviewer #3 (Recommendations For The Authors):**
1. Figures 2D and 6A are very unclear. Can the authors provide labels as text rather than coloured circles, whether the residue is on Q1 or E1? There is also a distance label in the figure in the small font with the faintest shade of grey, which I believe is supposed to be hydrogen bonds. Can this be improved for clarity?

We feel that additional labels on the ligand diagrams to be more confusing, instead, we updated the description in the legend and added labels to Figure 2B and Figure 6B to improve the clarity of residue positions. In addition, we have added 2 new figures with more detailed information about H-bonds (Figure 2-figure supplement 4, Figure 6- figure supplement 1).

1. Figure 2B - all side chains need labelling in different binding modes. The green ligand on blue protein is very difficult to see. Suddenly, the ligand turns light blue in panel 2C. Can this be consistent throughout the manuscript?

Figure 2B is updated according to this comment.

1. Figure 2 - figure supplement 2, and figure 6B. Can the author show the residue number on the x-axis instead of just the one-letter abbreviation? This requires the reader to count and is not helpful when we try to figure out where the residue is at a glance. I would suggest a structure label adjacent to the plot to show whether they are located with respect to the drug molecule.

Since the numbers for residues on either end of the cluster are indicated at the bottom of each boxed section, we feel that adding residue numbers would just further clutter the figure.

1. Figure 2 - figure supplement 2, and Figure 6B. Can you explain what is being shown in the error bar? I assume standard deviation?

Error bars on Figure 2-figure supplement 2 represent SEM. We added corresponding text in the figure legend.

1. Figure 2 - figure supplement 2, and figure 6B. Can you explain how many frames are being accounted for in this PBSA calculation?

For Figure 2- figure supplement 2 and Figure 6B a frame was made every 0.3 ns over 3x300 ns simulation, 1000 frames for each simulation, 3000 frames overall.

1. Figure 3D/E and 7C/D, it would be helpful to show which mutant show agreeable results with the simulations, PBSA/GBSA and contact analyses as suggested above.

The inconsistencies and discrepancies between the results of MD simulations and electrophysiological experiments are discussed throughout the manuscript.

1. Figure legend, figure 3E - I assume that there is a type that is different mutants with respect to those without the drug. Otherwise, how could WT, with respect to WT, has -105 mV dV1/2?

The reviewer is correct in that the bars indicate the difference in V1/2 between control and drug treatment. Thus, the difference in V1/2 (∆V1/2) between the V1/2 calculated for WT control and the V1/2 for mefenamic acid is indeed -105 mV. We have now revised Figure 3E's legend to accurately reflect this and ensure a clear understanding of the data presented.

1. Figure 3 - figure supplement 1B is very messy, and I could not extract the key point from it. Can this be plotted on a separate trace? At least 1 WT trace and one mutant trace, 1 with WT+drug and one mut+drug as four separate plots for clarity?

The key message of this figure is to illustrate the similarities of EQ WT + Mef and EQ L142C data. Thus, after thorough consideration, we have concluded that maintaining the current figure, which displays the progressive G-V curve shift in EQ WT and L142C in a superimposed manner, best illustrates the gradual shift in the G-V curves. This presentation allows for a clearer and more immediate comparison of the curve shifts, which may be more challenging to discern if the G-V curves were separated into individual figures. We believe that the existing format effectively communicates the relevant information in a comprehensive and accessible manner.

1. Figure 4B - the label Voltage is blended into the orange helix. Can the label be placed more neatly?

We altered the labels for this figure and added that information in the figure description.

1. Can you show the numerical label of the residue, at least only to the KCNE1 portion in Figures 4C and 4D?

We updated these figures and added residue numbering for clarity.

1. Can you hide all non-polar hydrogen atoms in figure 8 and colour each subunit so that it agrees with the rest of the manuscripts? Can you adjust the position of the side chain so that it is interpretable? Can you summarise this as a cartoon? For example, Q147 and Y148 are in grey and are very far hidden away. So as S298. Can you colour-code your label? The methionine (I assume M45) next to T327 is shown as the stick and is unlabelled. Maybe set the orthoscopic view, increase the lighting and rotate the figures in a more interpretable fashion?

We agree that Fig.8 is rather small as originally presented. We have tried to emphasize those residues we feel most critical to the study and inevitably that leads to de-emphasis of other, less important residues. As long as the figure is reproduced at sufficient size we feel that it has sufficient clarity for the purposes of the Discussion.

1. Line 538-539. Can you provide more detail on how the extracellular residues of KCNE3 are substituted? Did you use Modeller, SwissModel, or AlphaFold to substitute this region of the KCNEs?

We used ICM-pro to substitute extracellular residues of KCNE3 and create mutant variants of the Iks channel. This information is provided in the methods section now.

1. Line 551: The PIP2 density was solved using cryo-EM, not X-ray crystallography.

We corrected this.

1. Line 555: The system was equilibrated for ten ns. In which ensemble? Was there any restraint applied during the equilibration run? If yes, at what force constant?

The system was equilibrated in NVT and NPT ensembles with restraints. These details are added to methods. In the new simulations, we did equilibrations gradually releasing spatial from the backbone, sidechains, lipids, and ligands. A final 30 ns equilibration in the NPT ensemble was performed with restraint only for backbone atoms with a force constant of 50 kJ/mol/nm2. Methods were edited accordingly.

1. Line 557: Kelvin is a unit without a degree.

Corrected

1. Line 559: PME is an electrostatic algorithm, not a method.

Corrected

1. Line 566: Collecting 1000 snapshots at which intervals. Given your run are not equal in length, how can you ensure that these are representative snapshots?

Please see comment #5.

1. Table 3 - Why SD for computational data and SEM for experimental data?

There was no particular reason for using SD in some graphs. We used appropriate statistical tests to compare the groups where the difference was not obvious.